# CLOVER: CLOSED-LOOP VERIFIABLE CODE GENERATION

## ABSTRACT

The use of large language models for code generation is a rapidly developing trend in contemporary software development. However, without effective methods for ensuring the correctness of generated code, this trend could lead to any number of dangerous or even catastrophic outcomes. In this paper, we lay out a vision for addressing this challenge: the Clover paradigm, short for Closed-loop Verifiable Code Generation. At the core of Clover lies a checker that performs consistency checks among code, docstrings, and formal annotations. The checker is implemented using a novel integration of formal verification tools and large language models. We provide a theoretical analysis to support our thesis that Clover should be effective at checking the correctness of code. We also empirically investigate its feasibility on a hand-designed dataset (CloverBench) featuring annotated Dafny programs at a textbook level of difficulty. Experimental results show that for this dataset, (i) LLMs are reasonably successful at automatically generating formal specifications; and (ii) our consistency checker achieves a promising acceptance rate ($\geq 75\%$) for correct instances while maintaining zero tolerance for incorrect ones.

## 1 INTRODUCTION

Large language models (LLMs) have recently demonstrated remarkable capabilities. They can engage in conversation, retrieve and summarize vast amounts of information, generate and explain text and code, and much more (OpenAI, 2023; Chowdhery et al., 2022; Bubeck et al., 2023). Among many possible applications, their ability to synthesize code based on natural language descriptions (Chen et al., 2021; Li et al., 2022b; Cheng et al., 2023) is stunning and could potentially enhance the productivity of programmers significantly (Tabachnyk & Nikolov, 2022). Indeed, futurists are already claiming that in the future, most code will be generated by LLMs (or their successors) and not by humans.

However, there is a fundamental challenge that must be overcome before realizing any version of this future. Currently, there is no trustworthy way to ensure the correctness of AI-generated code (Liu et al., 2023). Without some quality control, the prospect of dramatically scaling up code generation is highly concerning and could lead to catastrophic outcomes resulting from faulty code Perry et al. (2022); Sandoval et al. (2023); Cotroneo et al. (2023). For the most part, the current best practice for curating AI-generated artifacts is to have a human expert in the loop (e.g., Github Copilot). While this is better than nothing, requiring human oversight of AI-generated code severely limits scalability. Furthermore, recent work (Pearce et al., 2022; Vaithilingam et al., 2022; Xu et al., 2022; Hendler, 2023) confirms the many risks and limitations of using AI even as a code assistant. Perhaps most concerning, results suggest that people with access to AI assistants write more insecure code, while at the same time having higher confidence in their code (Perry et al., 2022).

It is becoming clear that curating the quality of AI-generated content will be one of the most crucial research challenges in the coming years. However, in the specific case of generated code, there is a tantalizing potential solution. *Formal verification* can provide mathematically rigorous guarantees on the quality and correctness of arbitrary code. What if there were a way to *automatically* apply formal verification to generated code? This would not only provide a scalable solution, but it could actually lead to a future in which generated code is *more reliable* than human-written code.

Currently, formal verification is only possible with the aid of time-consuming human expertise. The main hypothesis of this paper is that *LLMs are well-positioned to generate the collateral needed to help formal verification succeed*; furthermore, they can do this *without compromising the formal guarantees provided by formal methods*. To understand how, consider the following breakdown of formal verification into three parts: (i) construct a mathematical model of the system to be verified; (ii) provide a formal specification of what the system should do; and (iii) prove that the model satisfies the specification. For code, step (i) is simply a matter of converting the code into mathematical logic, which can be done automatically based on the semantics of the programming language. And step (iii) can often be done automatically thanks to powerful automated reasoning systems for Boolean satisfiability (SAT) and satisfiability modulo theories (SMT) (Biere et al., 2009). In fact, a number of tools already exist that take the code and its specification (the result of step (ii)) as input and largely automate steps (i) and (iii) (e.g., Leino (2010); Barnes (2012)).[1] However, at first, step (ii) appears to be a showstopper for automated formal verification of generated code, as traditionally, significant human expertise is required to create formal specifications and ensure that they are both internally consistent and consistent with human expectations and understanding.

Two key insights suggest a way forward. The first insight is simply a shift in perspective: the result of any AI-based code generation technique should aim to include *not only code, but also formal specifications and natural language descriptions*. Our second insight is that given these components, we can use formal tools coupled with generative AI techniques to ensure that they are correct and consistent. We name our approach *Clover*, short for *Closed-loop Verifiable Code Generation*, and we predict that Clover, coupled with steadily improving generative AI and formal tools, will enable a future in which fully automatic, scalable generation of formally verified code is feasible. This paper charts the first steps towards realizing this vision.

The Clover paradigm consists of two phases. In the first phase, some process is used to create code annotated with a formal specification and accompanied by a natural language description (for simplicity, we refer to the latter two simply as "annotations" and "docstrings" going forward). This could be a one-shot generative process in which a generative AI agent creates all three parts. Alternatively, one or two of these components might already exist, in which case generative AI might be used to construct only the other(s). In fact, the second phase is completely agnostic to the process used in the first phase; we simply

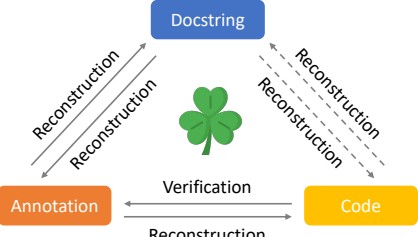

Figure 1: The Clover paradigm

insist that the result of the first phase has all three components: code, annotations, and docstrings. In the second phase, a series of *consistency checks* is applied to the code, annotations, and docstrings. Our claim is that if the consistency checks pass, then (i) the code is functionally correct with respect to its annotation; (ii) the annotation captures the full functionality of the code; and (iii) the docstring also accurately reflects the functionality of the code (see Figure 1). The idea is that we can unleash increasingly powerful and creative generative AI techniques in the first phase, and then use the second phase as a strong filter that only approves of code that is formally verified, well-documented, and internally consistent.

In this paper, we focus on the second phase, though we also include some demonstrations of the first phase in our evaluation. Our contributions include:

- a set of concrete steps for implementing the second phase (Section 3.1);
- a theoretical framework which argues for the trustworthiness of these steps (Section 3.2);
- the CloverBench dataset, featuring manually annotated Danfy programs with docstrings (Section 4.1);
- a feasibility demonstration of the first phase of the Clover paradigm using GPT-4, and an implementation and evaluation of the second phase of the Clover paradigm using GPT-4 and the Dafny verification tool (Section 4).

Our initial experiments on CloverBench are promising. Our implementation accepts 75% of the correct inputs and rejects 100% of the incorrect inputs. We expect that the acceptance rate can be improved in a variety of ways while maintaining the strong ability to reject incorrect code.

---

[1]While those tools have much room for improvement and will need to be retargeted to cover more mainstream languages, there are significant and separate research efforts in place to address this.

## 2 PRELIMINARIES: DEDUCTIVE PROGRAM VERIFICATION

Deductive program verification provides a framework for mathematically proving that programs are correct (Floyd, 1967; Hoare, 1969). The crux of deductive program verification is *annotating* code with preconditions, postconditions, and possibly loop invariants, and then checking that the code satisfies the specification given by these annotations. That is, if the code is executed starting from a program state that satisfies the precondition, the resulting program state after executing the code will satisfy the postcondition. Given such annotations, checking if the code satisfies them can be done by checking the validity of logical formulas known as verification conditions, which is typically done automatically using satisfiability modulo theories (SMT) solvers. Dafny is a state-of-the-art programming language and deductive verification tool (Leino, 2010), used in our evaluation.

In this paper, we assume annotations are given at the function level. For example, a function for finding the maximal element in an array of integers will have a precondition requiring that the input array is nonempty, and a postcondition ensuring that the return value is indeed the maximal element of the input array. Loops must be accompanied by loop invariants, which are used for a proof by induction on the number of loop iterations. For example, Listing 1 lists a Dafny function for finding the maximal element of an array, with a docstring, precondition, postcondition, and a loop invariant. Dafny is able to automatically verify that this function satisfies its annotation.

Listing 1: Dafny function with consistent code, annotation, and docstring.

```
// Find the maxiaml element in an integer array
method maxArray(a: array<int>) returns (m: int)
 requires a.Length>=1
 ensures exists k :: 0<=k<a.Length && m==a[k]
 ensures forall k :: 0<=k<a.Length ==> m>=a[k]
{
 m := a[0];
 var i := 1;
 while (i < a.Length)
 invariant 0<=i<=a.Length &&
           (forall k :: 0<=k<i ==> m>=a[k]) &&
           (exists k::0<=k<i && m==a[k])
 {
  m := if m>a[i] then  m else a[i];
  i := i + 1;
 }
}
```

Listing 2: Example of generated docstring.

```
// This method returns the maximum value, m,
// in the integer array a, ensuring that m
// is greater than or equal to all elements
// in a and that m is indeed an element of a
```

Listing 3: Example of generated annotation.

```
requires a.Length > 0;
ensures forall k::0<=k<a.Length ==> a[k]<=m
ensures exists k::0<=k<a.Length && a[k]==m
```

Listing 4: Example of generated code (loop invariant omitted).

```
var i := 0;
m := a[0];
while i<a.Length {
  if (a[i] > m) { m := a[i]; }
  i := i+1;
 }
```

## 3 CLOVER CONSISTENCY CHECKING

As mentioned in Section 1, the output of the first phase of the Clover paradigm must contain three components: code, annotation, and docstring. The Clover consistency checking algorithm checks the consistency between every pair of components, as shown in Figure 1, and accepts only if all checks accept. The docstring and annotation are consistent if they contain the same information, or they imply each other semantically. The notion of consistency between the docstring and code is similar. The consistency between the annotation and code is a bit different, as we can leverage deductive verification tools to check the soundness of the annotations formally. We explain how these checks are done in detail next. Detailed pseudocode is included in Appendix A.1 (Algorithm 1).

### 3.1 CONSISTENCY CHECKING ALGORITHM

Our key idea for checking the consistency between the three components depicted in Figure 1 is *reconstruction testing*. That is, given the three components (code, docstring, annotation) as input, we try to reconstruct a single component from a single other component, and then we check if the reconstructed result is equivalent to the original component. We do this for five out of the six (directed) edges of Figure 1. A special case is checking that the code conforms to the annotation, where we use formal verification based on deductive verification tools instead of a reconstruction test. For an input instance to pass the Clover consistency check, it must pass all six tests. For the reconstruction itself, we use an LLM (our evaluation uses GPT-4), and for equivalence testing,

we use LLMs to compare text, formal tools to compare annotations, and pointwise sampling to compare code. More details are provided below, and a running example is provided in the appendix (Section A.4). Listing 2, Listing 3 and Listing 4 are examples of generated artifacts.

**Code-Annotation Consistency** (1. Code → Annotation: Soundness) A deductive verification tool (our evaluation uses Dafny) checks that the code satisfies the annotation. This is a standard formal verification check (see Section 2), and it is sound in the sense that it will never pass if the code is inconsistent with the annotation. (2. Annotation → Code: Completeness) To prevent an annotation that is too trivial from being accepted, we test if the annotation is strong enough by testing if it contains enough information to reconstruct functionally equivalent code. Given the annotation, we use an LLM to generate new code. Then, we check the equivalence between the generated and the original code. If the equivalence check passes, the annotation is considered complete.

**Annotation-Docstring Consistency** (1. Annotation → Docstring) An LLM is asked to generate a new docstring from the annotation. Then, the new and the original docstrings are checked for semantic equivalence. (2. Docstring → Annotation) An LLM is asked to generate a new annotation from the docstring. Then, the new and the original annotations are checked for logical equivalence.

**Code-Docstring Consistency** (1. Docstring → Code) An LLM is asked to generate code from the docstring. Then, the new and the original code are checked for functional equivalence. (2. Code → Docstring) An LLM is asked to generate a new docstring from the code. Then, the new and the original docstrings are checked for semantic equivalence.

We consider the methods used for equivalence checking as parameters of the Clover paradigm. We discuss some possibilities (including those used in our evaluation) below.

**Equivalence Check for Code** Standard equivalence tests for code include using input-output comparisons, concolic testing (King (1976); Cadar et al. (2008); Cadar & Sen (2013); Udupa et al. (2013)), or even full formal equivalence checking (e.g. Churchill et al. (2019)). Our evaluation uses a set of input-output pairs that are included as part of our dataset CloverBench. This test is, of course, imprecise (complete but not sound), but our evaluation suggests that it suffices for the level of complexity in CloverBench. More advanced equivalence checking techniques might be required for more complex examples. For example, the generated code of Listing 4 is equivalent to the original code in Listing 1 and indeed Clover accepts it as so.

**Equivalence Check for Docstrings** Checking equivalence between docstrings is challenging, as natural language is not mathematically precise. In our evaluation, we ask an LLM (GPT-4) to check whether two docstrings are semantically equivalent. For example, it accepts Listing 2 as equivalent to the docstring in Listing 1. Other NLP-based semantic comparisons may also be worth exploring.

**Equivalence Check for Annotations** To check the equivalence of two annotations, we write the equivalence of the two annotations as a formal lemma and ask a formal tool (in our evaluation, we again use Dafny) to prove the lemma. This method is conservative in the sense that it succeeds only if the two annotations are indeed equivalent, but it may fail on equivalent annotations due to limitations of the verification tool being used. For example, we are able to automatically prove that the annotation in Listing 3 is equivalent to the one in Listing 1. The specific equivalence checking template we use is described in Section 4.1 and is included as part of our dataset CloverBench.

## 3.2    AN ANALYTICAL MODEL OF CLOVER

This section provides a mathematical rationale and analysis for Clover's reconstruction tests. For the purpose of the analysis we focus on a single directed edge from domain $A$ to domain $B$ (e.g., code to docstring). The full Clover test uses five instances of this reconstruction test (see Figure 1).

We assume each domain is equipped with a semantic equivalence relation, denoted by $\equiv$. Each domain can therefore be partitioned into equivalence classes. For $X \in \{A, B\}$ we use $e(X)$ to denote the set of equivalence classes of $X$, and for $x \in X$ we use $[x]$ to denote the equivalence class $x$ belongs to. For docstrings, the equivalence relation represents two docstrings being semantically equivalent (e.g. under human understanding of the docstrings); for code, equivalence represents functional equivalence; and for annotations equivalence represents logical equivalence.

We further assume a *ground truth consistency relation* between $A$ and $B$, denoted by $G \subseteq A \times B$. The ground truth consistency represents the consistency we assume to exist between docstrings,

annotations, and code, as described in Section 3.1. We assume the consistency relation satisfies the following properties that link it to the equivalence relation: For any $x, x' \in A$ and $y, y' \in B$, $(x \equiv x' \wedge y \equiv y') \rightarrow ((x, y) \in G \leftrightarrow (x', y') \in G)$ and $((x, y) \in G \wedge (x', y') \in G) \rightarrow (x \equiv x' \leftrightarrow y \equiv y')$. That is, consistency is preserved when substituting equivalent objects, and any object may be consistent with at most one equivalence class from the other domain.

We now formally define and analyze the *single-edge Clover consistency test*, which aims to be an approximate test for $G$. For the analysis, we assume a probability distribution $D$ on $A \times B$. The test relies on a *transfer model* and the analysis assumes it is *transfer-rational*, as defined below.

**Definition 3.1** (Transfer Model)**.** Given domains $A$ and $B$, a transfer model for $A$ and $B$ is a function $M : A \times B \to \mathbb{R}$ such that for each $x \in A$, $M(x, \cdot)$ is a probability distribution over $B$. Here $M(x, y)$ denotes the probability of transferring $x \in A$ to $y \in B$.

**Definition 3.2** (Transfer-Rational Model)**.** Let $M$ be a transfer model for $A$ and $B$. We say $M$ is *transfer-rational* if for each $x \in A$ there is a unique $[y] \in e(B)$ that maximizes $\sum_{y' \in [y]} M(x, y')$. In this case, we define the *transfer function of $M$*, $f^M : A \to e(B) = \lambda x. \arg \max_{[y] \in e(B)} \sum_{y' \in [y]} M(x, y')$.

Intuitively, the transfer model is meant to approximate a mapping based on the ground truth consistency ($G$). In the context of Clover, the domains are among docstring, annotation, and code, and the transfer model is given by an LLM (GPT-4). For example, when $A$ is docstrings and $B$ is annotations, the distribution $M(x, \cdot)$ represents the output distribution of GPT-4 on an input docstring $x$ with a suitable prompt for generating an annotation corresponding to the docstring $x$.

We now fix a transfer-rational model $M$, and define the single-edge Clover consistency check.

**Definition 3.3** (Single-Edge Clover Consistency Check)**.** For input $x \in A, y \in B$, the single-edge Clover consistency check (for the edge from $A$ to $B$) is a procedure that draws $y'$ from the distribution $M(x, \cdot)$, and then accepts if $y' \equiv y$ and otherwise rejects.

Note that the check relies on being able to check equivalence in domain $B$.[2]

We now analyze the probability that the single-edge Clover consistency check is correct. Our analysis relies on two assumptions: one relating the transfer model $M$ with the ground truth consistency $G$, and another ensuring that $M$'s distributions are concentrated.

**Assumption 3.4** (Consistency Alignment)**.** Let $c_1$ be the probability that $y \in f^M(x)$ when $x, y$ are sampled from $A \times B$ according to $\mathcal{D}$ conditioned on $(x, y) \in G$. Similarly, let $c_0$ be the probability that $y \in f^M(x)$ when $x, y$ are sampled from $A \times B$ according to $\mathcal{D}$ conditioned on $(x, y) \notin G$. We assume that $c_1$ is close to 1, and $c_0$ is close to 0.

**Assumption 3.5** (Concentration)**.** Consider $x, y$ sampled from $A \times B$ according to $\mathcal{D}$ conditioned on $(x, y) \in G$ and $y \in f^M(x)$. We assume that for some non-negligible $0 < l \leq 1$ (e.g. 30%), the following holds with probability $\geq p_c$ ($p_c$ close to 1): $\sum_{y \in f^M(x)} M(x, y) \geq l$. Similarly, consider $x, y$ sampled from $A \times B$ according to $\mathcal{D}$ conditioned on $(x, y) \notin G$ and $y \notin f^M(x)$. We assume that for some $u$ significantly smaller than $l$, the following holds with probability $\geq p_c$: $\max_{[y] \in e(Y), [y] \neq f^M(x)} \sum_{y' \in [y]} M(x, y') \leq u$.

Intuitively, the concentration assumption means that with high probability ($\geq p_c$), sampling from $M$ is the same as applying $f^M$, and specifically that the second most likely equivalence class is much less likely than the maximal one (i.e., the one given by $f^M$).

**Theorem 3.6.** *Under Assumptions 3.4 3.5, consider $(x, y)$ sampled from $A \times B$ according to $\mathcal{D}$ conditioned on $(x, y) \in G$; the single-edge Clover consistency check will accept $(x, y)$ with probability $A \geq l \cdot p_c \cdot c_1$. Similarly, consider $(x, y)$ sampled from $A \times B$ according to $\mathcal{D}$ conditioned on $(x, y) \notin G$; the single-edge Clover consistency check will will accept with probability $R \leq u \cdot p_c \cdot (1 - c_0) + (1 - p_c)(1 - c_0) + c_0$.*

Theorem 3.6 ensures that under our assumptions, the probability of accepting a consistent input is significant, and the probability of accepting an inconsistent input is negligible. We can increase the

---

[2]We assume a perfect equivalence check to keep the analysis simple and illustrative. In practice, the equivalence tests do incur some imprecision. But accounting for this imprecision using a probabilistic model is cumbersome because the distribution on the equivalence checks Clover performs depends on both the input distribution and on the transfer model.

gap by repeating the reconstruction test several times (but not too many) and accepting if any of them accept. As discussed in Section 4, our evaluation uses 3 reconstruction attempts.

**From the single-edge to the full Clover consistency check.** Our analysis focused on a single, directed reconstruction edge (see Figure 1), while the full Clover consistency check uses five reconstruction edges and a single verification edge, and accepts only if all six checks accept. We do not attempt to theoretically analyze the full check, because we do not assume the edges to be independent (so multiplying acceptance probabilities is not necessarily meaningful). In our experiments we empirically measure the acceptance rate of each edge, and also observe that the edges are not independent (see Section 4.5).

While in this section we used an edge to denote a single directed edge, in the sequel we sometimes use an edge to refer to the combined check of both directions of an edge from Figure 1.

## 4    EVALUATION

We have implemented a first prototype of our Clover consistency checking algorithm using GPT-4 (OpenAI, 2023) as the LLM and using the Dafny programming language and verification tool (Leino, 2010). We selected Dafny because it provides a full-featured and automatic deductive verification toolkit including support for a rich language of formal specifications and a backend compiler linking to a verifier. But of course, Clover can be instantiated using any language and tool supporting deductive program verification. Note that it is also crucial that the selected LLM has a good understanding of the programming language used. In our case, we were pleasantly surprised to discover that GPT-4 understands Dafny programs well enough to perform the tranlations between code, docstrings, and annotations that Clover relies on, despite the fact that Dafny is not a mainstream programming language. In our evaluation, we use Dafny version is 4.0.0.50303 with Z3 version 4.8.12. The evaluation also uses a concrete set of Dafny examples which we describe next.

### 4.1    DATASET: CLOVERBENCH

There have been several popular datasets for code generation in different domains (Austin et al., 2021; Chen et al., 2021; Hendrycks et al., 2021; Yin et al., 2023; Lai et al., 2023), but none contain annotations or use the Dafny language. Furthermore, we wanted to carefully curate the programs used to test our first Clover prototype. For these reasons, we introduce a new hand-crafted dataset we call CloverBench. We expect to add and improve it over time, but at the time of writing, it is based on 60 small hand-written example programs as might be found in standard CS textbooks.[3] For each program, there are four variants: a "ground-truth" variant whose code, annotation, and docstring are correct and consistent (verified by hand); and 3 incorrect variants.Associated with each example, there is also one set of unit tests and one Dafny code template for annotation equivalence checking.

**Unit Tests** Each set of unit tests contains five individual tests designed for each example. We use this dataset to determine if a piece of generated code is equivalent to the original code. If the generated code passes all five tests, then the code is considered to be equivalent. We run the tests by using Dafny with the compiler enabled and the verifier disabled.[4]

**Annotation Equivalence Checking Template** Each template can be used to formally verify the consistency between two annotations with Dafny. For two annotations $a$ and $b$ to be equivalent, the preconditions and postconditions of $a$ and $b$ must be verified to be equivalent separately. Details and an example are shown in the appendix A.5.

### 4.2    PHASE 1: GENERATION

Because Clover relies on phase one being able to produce code with annotations and docstrings, our first experiment explores the feasibility of this assumption. In particular, we test GPT-4's ability to generate annotations and code using different configurations. Figure 2a shows the results when GPT-4 is asked to generate the code for each of the 60 examples in CloverBench under various

---

[3]Since we wanted to concentrate on the most basic scenario initially, our initial dataset only features examples containing exactly one method and no helper functions or methods.

[4]This only applies to equivalence checks. Of course, the verifier *is* enabled when doing formal checks.

conditions. The first bar ("one try") shows the result when asking GPT-4 to produce the code, given the annotation, in a single try. The next bar allows GPT-4 to try three times, each time providing the output of the Dafny compiler and verifier as feedback. The next is similar, but using the output of only the Dafny compiler. In the last bar, we allow three tries with feedback from the Dafny compiler and verifier and also provide the docstring. We see that, at its best, GPT-4 can correctly provide the code for 53 out of 60 examples, and it does best when it gets the most feedback from Dafny.

Figure 2b shows results from asking GPT-4 to generate annotations when provided just the code (here, annotations include pre- and post-conditions and loop invariants). In one try, GPT-4 succeeds on 28 of 60 programs. Given 3 tries and maximal feedback from Dafny, this improves to 41 of 60. Though not perfect, out of the box, GPT-4 can produce correct annotations for the majority of programs in our simple set of benchmarks. This suggests that using LLMs for specification generation is feasible, and further efforts in this direction (including fine-tuning models for the task) are likely to lead to even stronger capabilities.

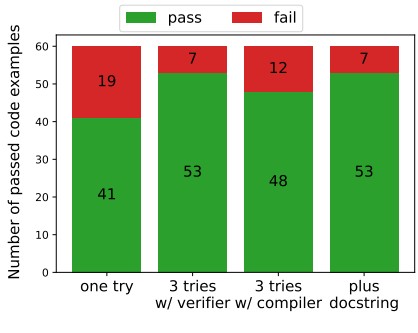

(a) Code generation.

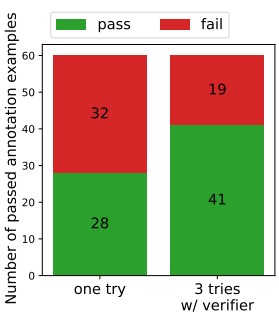

(b) Annotation generation

Figure 2: Phase one feasibility study

## 4.3 PHASE 2: RESULTS ON GROUND-TRUTH EXAMPLES

Our main experiment evaluates the capabilities of the Clover consistency checking algorithm.[5] For each example, we run all 6 checks described in Section 3.1. The end-to-end results are summarized in Table 1. We see that Clover accepts 45 of 60 correct ("ground truth") examples and rejects all incorrect examples. Details on each of the 6 checks for the ground truth examples is shown in Table 2. All acceptance rates are above $80\%$. Failures are mostly due to incorrect or imprecise reconstruction. Generation of invalid Dafny syntax is the most common reason for failure. More details can be found in Appendix A.2.5. We expect that using better LLMs (either better general-purpose LLMs or LLMs fine-tuned for program verification or a specific language or both) will improve the acceptance rate. More details and a running example see Section A.4 in the Appendix.

Since ground-truth examples are hand-written and checked for correctness, all pass the Dafny verifier (i.e., all annotations are sound). Annotation completeness requires successful synthesis from annotation to code, and here, we get an $88\%$ acceptance rate. The main failure reason is incorrectly generated Dafny syntax by GPT-4. In doc2anno generation, we generate annotations from docstrings. The main failure comes again from GPT-4 generating incorrect Dafny syntax. anno2doc and code2doc have almost perfect acceptance rates. On the one hand, this is because GPT-4 is very good at synthesizing natural language. On the other hand, our docstring equivalence checker is not very strong and skews towards acceptance. As long as they don't directly contradict each other, information omissions or additions in docstrings frequently go unnoticed by GPT-4. Improving this equivalence checker is one important direction for future work. doc2code generation shares the same issues as anno-complete and doc2anno: failure because of invalid Dafny syntax generation.

---

[5]During consistency checking, we consider everything that appears in the body of a function, including assertions and invariants, to be part of the code, and consider the annotation to consist only of pre- and post-conditions.

|  | Accept |
|---|---|
| Correct | 45/60 (75%) |
| Wrong-C1 | 0/60 (0%) |
| Wrong-C2 | 0/60 (0%) |
| Wrong-C4 | 0/60 (0%) |

Table 1: Summary of phase 2 experiments.

| Correct | Accept |
|---|---|
| anno-sound | 60/60 (100%) |
| anno-complete | 53/60 (88%) |
| doc2anno | 51/60 (85%) |
| anno2doc | 60/60 (100%) |
| code2doc | 58/60 (97%) |
| doc2code | 49/60 (82%) |

Table 2: Ground-truth acceptance.

## 4.4 PHASE 2: RESULTS ON INCORRECT EXAMPLES

As mentioned, for each program in our dataset, we created 3 incorrect versions. Here we describe them in more detail. Table 3 lists four categories of incorrect programs. Categories C1, C2, and C3 contain programs in which exactly one component is incorrect and the other two are correct. We note that C3, in which the code is functionally incorrect but the annotation is correct, is trivial because Dafny will always fail to verify the code in this situation. If two components are wrong, the most interesting case is when the annotation and code are consistent and thus not detectable by Dafny. Category C4 captures this situation. Our incorrect examples are drawn from C1, C2, and C4.

| C1 | Annotation wrong, docstring and code consistent. |
|---|---|
| C2 | Docstring wrong, annotation and code consistent. |
| C3 | Code wrong, annotation and docstring consistent. This case is trivial because the example will always fail with Dafny. |
| C4 | Docstring correct, annotation and code consistent but wrong. |

Table 3: Incorrect sample categories

Each category has sub-categories, and we construct incorrect examples in such a way as to have the same number in each sub-category. Table 6 shows the different sub-categories of C1. For each, we show whether Dafny will verify the example. The two most interesting cases are when Dafny verification is uncertain. We draw our mutated examples from these two sub-categories. For C2, the problem can either be that the docstring is too strong (contains more information than necessary), the docstring is wrong (contains information conflicting with code or annotation), or the docstring too weak (omits some information contained in the code and annotation). For C4, the problem can either be code-annotation intention error or the code is wrong and the annotation is too weak to detect it. Table 4 shows the results of the 6 tests for each category. We observe that doc2anno has the highest rejection rate. This is because annotation equivalence checking uses formal checks with Dafny and can guarantee soundness.

## 4.5 REVISITING THE ANALYTICAL MODEL

Here, we empirically estimate the values of $A$ and $R$ from Theorem 3.6 based on our experiments. That is, we estimate the acceptance rate for correct and incorrect inputs for each directed edge. Each cell in Table 5 represents the percentage of reconstructed components that successfully pass the equivalence check in the four categories: ground-truth (Table 2), C1, C2, and C4 (Table 3).[6]

| Edge | C1 Reject | C2 Reject | C4 Reject |
|---|---|---|---|
| anno- sound | 32/60 (53%) | 0/60 (0%) | 0/60 (0%) |
| anno- complete | 25/60 (42%) | 8/60 (13%) | 47/60 (78%) |
| doc2anno | 60/60 (100%) | 57/60 (95%) | 60/60 (100%) |
| anno2doc | 31/60 (52%) | 44/60 (73%) | 42/60 (70%) |
| code2doc | 2/60 (3%) | 49/60 (82%) | 43/60 (72%) |
| doc2code | 11/60 (18%) | 32/60 (53%) | 52/60 (87%) |

Table 4: Rejection rates for incorrect examples.

---

[6]Note that our incorrect examples are constructed with the aim of making them hard to reject, i.e., by considering only the cases that can pass Dafny verification. The measured values for $u$ are thus likely to be higher than the value for a more natural distribution.

As mentioned above, in the first column, the discrepancy between the measured acceptance rate and the ideal perfect acceptance rate comes partly from reconstruction failures and partly from equivalence checker failures. For example, the doc2anno acceptance rate is 0.85, not 1. Apart from the failure to generate the correct annotation, there are also cases where the generated annotation is correct but unable to be verified by our annotation equivalence checking template in Section 4.1.

Overall, the measured aggregated acceptance rate for the first column is 0.75. This is higher than would be expected if each check were independent (the product of the entire column is 0.59). This is because in practice, they are not independent: easier examples that pass the tests on one edge tend to also pass the tests on other edges. In C1, anno-sound and doc2anno both have zero acceptance rate for inconsistent edges, and the overall acceptance rate is also zero. In C2, none of the edges have zero acceptance but the intersection of acceptance is empty. In C4, doc2anno has a zero acceptance rate, and the overall acceptance is also zero. Note that the anno2doc acceptance rate is high for C1, C2, and C4. This is because our current docstring equivalence checker is good at detecting contradictory information but not the addition or omission of information due to a slightly strengthened or weakened annotation.

| edge | Accept Correct $A$ | Accept Incorrect $R^{C1}$ | Accept Incorrect $R^{C2}$ | Accept Incorrect $R^{C4}$ |
|---|---|---|---|---|
| anno- sound | 1 | 0 | − | − |
| anno- complete | 0.88 | − | − | 0.22 |
| doc2anno | 0.85 | 0 | 0.05 | 0 |
| anno2doc | 1 | 0.48 | 0.27 | 0.30 |
| code2doc | 0.97 | − | 0.18 | 0.28 |
| doc2code | 0.82 | − | − | 0.13 |

Table 5: Empirically measured values for $A$ and $R$. Entries shown as "−" are omitted because for that category and check, the assumptions of the analytical model are violated. Here the value of anno- sound for the C1 columns are taken from a subset of C1, because the other part are designed to pass the anno- sound test.

## 5 RELATED WORK

Beside the notable (Chen et al., 2021; Li et al., 2022b; Cheng et al., 2023) for code generation using LLMs, Gulwani et al. (2017) is a survey on program synthesis before the era of LLMs. And other works using neural approaches for program synthesis (Yin & Neubig, 2017; Austin et al., 2021; Bowers et al., 2023). To scale up the generation, there have been attempts to decompose the whole task into smaller steps (Zelikman et al., 2023; Ellis et al., 2021; Bowers et al., 2023) and also to use execution traces (Shi et al., 2022b; Ding et al., 2023). While the aforementioned work synthesize code from natural language, another common theme is to synthesize programs from specifications (Alur et al., 2013; Polikarpova et al., 2016; Solar-Lezama, 2008; Chaudhuri et al., 2021). Translation between natural languages and formal languages has been studied in Hahn et al. (2022); Sun et al. (2023); Ghosh et al. (2022), and and Pei et al. (2023) studies how to use LLMs for predicting program invariants. Verifying if a generated program is correct is challenging. In Liu et al. (2023), the test-case approach to check code correctness is demonstrated to be insufficient. Other attempts include Key et al. (2022), which ask the model to generate assertions along with the code, and Chen et al. (2021) Chen et al. (2023) which study generation of unit tests. There is also a line of works Cobbe et al. (2021); Li et al. (2022a); Zhang et al. (2023); Inala et al. (2022) on the learning-based approach for verifying correctness. Shi et al. (2022a); Li et al. (2022b); Inala et al. (2022); Wei et al. (2022) have studied various approaches for reranking the model's output. Charalambous et al. (2023) proposed a self-repair method combining LLMs and formal verification strategies.

## 6 CONCLUSION

We have introduced Clover, a framework for closed-loop verifiable code generation. Initial experiments using GPT-4, Dafny, and a set of simple textbook examples are promising. There are many avenues for future work, including: better verification tools, improving LLM capabilities for generating code, annotations, and docstrings, improving LLM capabilities for understanding Dafny syntax, and scaling up to more challenging examples.

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

# A  APPENDIX

## A.1  PSEUDOCODE OF CLOVER CONSISTENCY CHECK

Algorithm 1 is the pseudocode of Clover consistency check.

---

**Algorithm 1** Clover Consistency Check

---

**Input:** Docstring $d$, annotation $a$, and code $c$.
**Output:** $True/False$
  **if** Dafny failed to verify $a, c$ **then**
    **Return** $False$

  Set number of tries $k = 3$
  $p = False$
  **for** $i = 1$ to $k$ **do**
    Call GPT-4 to generate code $c'$ from $a$.
    **if** $c'$ is equivalent to $c$ **then**
      $p = True$
  **if** $p == False$ **then**
    **Return** $False$

  $p_1 = p_2 = False$
  **for** $i = 1$ to $k$ **do**
    Call GPT-4 to generate code $c'$ from $d$.
    **if** $c'$ is equivalent to $c$ **then**
      $p_1 = True$
  **for** $i = 1$ to $k$ **do**
    Call GPT-4 to generate docstring $d'$ from $c$.
    **if** $d'$ is equivalent to $d$ **then**
      $p_2 = True$
  **if** $p_1 == False$ or $p_2 == False$ **then**
    **Return** $False$

  $p_1 = p_2 = False$
  **for** $i = 1$ to $k$ **do**
    Call GPT-4 to generate annotation $a'$ from $d$.
    **if** $a'$ is equivalent to $a$ **then**
      $p_1 = True$
  **for** $i = 1$ to $k$ **do**
    Call GPT-4 to generate docstring $d'$ from $a$.
    **if** $d'$ is equivalent to $d$ **then**
      $p_2 = True$
  **if** $p_1 == False$ or $p_2 == False$ **then**
    **Return** $False$

---

## A.2  DISCUSSIONS

### A.2.1  LIMITATIONS

There are many limitations in the proposed schema as the GPT-4's capability is limited. The docstring, annotation, and code loop also have inherent limitations. For example, Annotations cannot be complete and are designed to hide implementations sometimes. It can describe some input/output desired properties but cannot force the implementation. E.g., it can force an array to be sorted but cannot restrict the algorithm used for sorting. In this paper, as a first step, we only aim to check the functional consistency (correctness) but doesn't guarantee the performance of the implementation.

As mentioned in Section 3, if the oracle used for consistency check has an unaligned mapping with humans, e.g., understand a sorting algorithm as getting maximum value, there is no way to examine it without human intervention. But in practice, we think this will happen rarely, as the Assumption 3.4. On the other hand, if all the docstring, annotation, and code miss the same edge case, the error can never be detected. This is correct by definition as they are consistent but may not follow the best code practice. We haven't detected such errors in our experiments, but it can happen. For Clover to be used in production in the future, we still expect there will be a real scenario test or human interaction to cover the tail cases.

### A.2.2  CLOVER VARIANTS

In Clover, three components need to be checked for consistency. While the triangle in Clover is not the unique pattern for consistency check, we'd like to discuss two other variants.

Currently, a common pattern of code generation includes only docstring and implementation. However, a closed loop between docstring and code cannot ensure implementation correctness, as docstring is not a precise representation. The consistency check between docstring and implementation can check the human intention following, but it is hard to detect subtle implementation errors.

Another interesting variant is to put unit tests in the loop. This is a common way of evaluating the generation accuracy on many coding benchmarks like in MBPP and HumanEval (Austin et al.

(2021); Chen et al. (2021)). We also think it will be great to add unit tests into Clover, and we left this as a future work. While we believe unit tests could be very helpful in some cases, they also have weaknesses. On the one hand, it could sometimes be harder to generate than annotations. The unit test could be unexplainable, making it hard to be trusted. Also, if a model can generate effective unit tests, that means the model can predict the execution at some extent. Execution can take many calculation steps, which is not a solved task for LLM. On the other hand, unit tests are not a strong guarantee of correctness compared to annotation.

### A.2.3 FUTURE RESEARCH

A successful Clover paradigm relies on many components. To unlock the full potential of Clover, several downbreak topics can benefit it. Each of them can be pushed independently and has a broader use case. The three basic components are high-quality code generation, annotation generation, and docstring generation. They will affect the success rate of Clover on correct generations and make Clover realistic in practice. If we have high-quality unit-test generation, then the unit-test can be added into Clover. To have a reliable consistency check, despite the generation capability, the equivalence check is also a critical part. The equivalence check for docstring could be a problem for LLM. The equivalence check for annotation is currently handled by Dafny, which also related to theorem proving. For code, equivalence check is a long-lasting problem, while fuzzing and concolic testing are two mainstream techniques.

### A.2.4 DATA CONTAMINATION

We'd like to point out that the current version of CloverBench has some limitations. We hand-crafted it starting with textbook-level examples because the GPT-4's ability on Dafny generation is limited. But then it comes with the issue of data contamination. Our examples may not be directly in the training data, but with significant probability, GPT-4 has seen something similar before. Even if only the code with similar texture in other languages has been seen in the training data, our hand-crafted examples can be affected. Some soft evidence is that the GPT-4 can generate the correct code sometimes even with incomplete docstring and incomplete annotation. We observed that sometimes just the function signature is good enough, so we tried to avoid such an issue in our experiments by replacing the function name with non-meaningful names. In future works, we plan to fine-tune models for program verification semantics and Dafny. After the model's capability of Dafny increases, we'll update CloverBench with harder examples.

### A.2.5 FAILURE REASONS OF GPT-4

From our observation, GPT-4 is not trained on the current version $4.0.0$ of Dafny and, therefore, very bad at its syntax. One can imagine with an LLM trained or fine-tuned on Dafny $4.0.0$, we can easily get a much higher acceptance rate. Here is some evidence that GPT-4 is not at all familiar with the current Dafny syntax. Annotations require `modifies clause` or `reads clause` when methods want to access memory. So `reads array` in the annotation is needed when we are reading an array element, and GPT-4 misses it almost $100\%$ of the time at its first try at generation. Luckily, with Dafny compiler-generated error messages, GPT-4 is often able to add the needed `modifies clause` or `reads clause`. Another example is that Dafny used to require annotations to be separated by semicolons, or assert explicitly that an array is not null `requires array!=null` in the pre-conditions but not anymore. But GPT-4 still adheres to those deprecated rules. For a complete view of our GPT-4 system prompts, see A.5.

### A.3 PROOF OF THEOREM 3.6

1. For $(x, y)$ sampled from $\mathcal{D}$ with the condition $(x, y) \in G$. From Assumption 3.4, with probability $\geq c_1$, we have $y \in f^M x$. Then, according to Assumption 3.5 and the perfect equivalence oracle, with probability $p_c$, the reconstruction from $x$ to $y$ will succeed with probability $\geq l$. Therefore, the accept probability is $\geq l \cdot p_c \cdot c_1$, denoted as $A$.
2. For $(x, y)$ sampled from $\mathcal{D}$ with the condition $(x, y) \notin G$. There are 3 cases:
   – From Assumption 3.4, with probability $c_0$, there is $y \in f^M(x)$, and it is trivial that the accept probability $\leq 1$.

- With probability $1 - c_0$, there is $y \notin f^M(x)$. Then from Assumption 3.5, with probability $p_c$, the reconstruction from $x$ to $y$ will succeed with probability $\leq u$.
- Finally, the last case that the bounds in Assumption 3.5 does not hold, with the rest probability $(1 - c_0)(1 - p_c)$, it is trivial that the accept probability $\leq 1$.

By aggregating all the cases, the accept probability is $\leq c_0 + (1 - c_0) \cdot p_c \cdot u + (1 - c_0)(1 - p_c)$.

### A.4 EXAMPLE OF EACH CONSISTENCY CHECK

We describe each experimental step and use `maxArray` as a running example.

Listing 5: Annotation Input

```
method foo(a: array<int>) returns (m: int)
 requires a.Length >= 1
 ensures (forall k :: 0<=k<a.Length ==> m>=a[k]) && (exists k :: 0<=k<a.Length && m==a[k])
{
//TOFILL
}
```

Listing 6: Code Input

```
method foo(a: array<int>) returns (m: int)
//TOFILL
{
 m := a[0];
 var i := 1;
 while (i < a.Length)
 invariant 0<=i<=a.Length &&
           (forall k :: 0<=k<i ==> m>=a[k])
     &&
           (exists k :: 0<=k<i && m==a[k])
 {
  m := if m>a[i] then  m else a[i];
  i := i + 1;
 }
}
```

Listing 7: Generated code

```
method foo(a: array<int>) returns (m: int)
{
  var i := 0;
  m := a[0];
  while i<a.Length
  {
    if(a[i] > m) {
      m := a[i];
    }
    i := i+1;
  }
}
```

Listing 8: Docstring Input

```
// specification: Returns the maximum value m present in the array a.
method maxArray(a: array<int>) returns (m: int)
//TOFILL
```

**anno-sound** Annotation soundness can be checked directly with Dafny verifier.

**anno-complete** For the annotation to be complete with respect to the code, we must be able to reconstruct the code from the annotation alone. Therefore, we use GPT-4 to generate code from the masked function signature and the annotation (Listing 5) with Dafny verifier-generated feedback up to three tries to help GPT-4 fix its code generation (Listing 7). Then we check if the generated code is equivalent to the original ground-truth code with unit tests described in Section 4.1.

**doc2anno** We try to reconstruct an equivalent ground-truth annotation from the docstring alone. First, we call GPT-4 with the docstring and the function signature (Listing 8) asking for the annotation (Listing 9). To eliminate simple syntax errors, we try to compile the generated annotation with an empty code body and feedback Dafny compiler-generated error messages. From our ablation studies, we discover that the feedback mechanism is necessary and very effective as described in 4.2.

Listing 9: Generated annotation

```
  requires a.Length > 0;
  ensures forall k :: 0 <= k < a.Length ==> a[k] <= m;
  ensures exists k :: 0 <= k < a.Length && a[k] == m;
```

**anno2doc** First, we try to recreate an equivalent docstring (Listing 10) from the annotation. For two docstrings to be equivalent, they must contain the exact same information about the functional behavior of a program and include no implementation details. Then, to check if the two docstrings are semantically equivalent, we again call GPT-4 to see if the two docstrings describe the same

annotation (Prompt in A.5). Note that the two calls to GPT-4 are independent to ensure that the second call contains no memory of the first call. As a result, the answer to the question of whether the original and the generated docstrings are semantically equivalent is unaffected by the first call to generate an equivalent docstring from the original.

Listing 10: Generated docstring

```
'''This method returns the maximum value, m, in the integer array a, ensuring that m is
    greater than or equal to all elements in a and that m is indeed an element of a.'''
```

**code2doc** The process is almost identical to `anno2doc`. The only difference is that to ensure the code provides all the information needed for the docstring generation, we put the preconditions in the code in the form of `assert statement;`.

**doc2code** The process is one of the most common use cases of GPT-4: generating the code from a natural language description. The concrete execution is similar to that described in `anno-complete`. The only difference is that instead of using verifier-generated error messages, we use compiler-generated error messages since we want to ensure that the code generation strictly comes from the docstrings.

## A.5 SUPPLIMENTARY TEMPLATE AND EXAMPLES

In Listing 12, we give a template for verifying annotation equivalence for the ground-truth example `max_array`11. Verification is done by verifying the template with Dafny's auto-verifier. If the lemmas `pre_eq` and `post_eq` are both verified, then it means that Dafny has successfully verified the equivalence of pre- and postconditions. `predicate pre_original` states the full preconditions of the ground-truth example and `predicate post_original` states the full postconditions. `predicate pre_gen` will be replaced by the generated preconditions and `predicate post_gen` will be replaced by the generated postconditions. The lemma `pre_eq` states that the generated preconditions are true if and only if the original preconditions are true. The lemma `post_eq` states that the generated postconditions are true if and only if the original postconditions are true. The above example is simple enough to be proven by the auto-prover.

Note that Dafny is sound but not complete, that is, there could be cases when two predicates are indeed functionally equivalent but Dafny cannot prove it. An exmaple is shown in Listing 13.

Listing 11: maxArray

```
method maxArray(a: array<int>) returns (m: int)
  requires a.Length >= 1
  ensures forall k :: 0 <= k < a.Length ==> m >= a[k]
  ensures exists k :: 0 <= k < a.Length && m == a[k]
{
  m := a[0];
  var index := 1;
  while (index < a.Length)
    invariant 0 <= index <= a.Length
    invariant forall k :: 0 <= k < index ==> m >= a[k];
    invariant exists k :: 0 <= k < index && m == a[k];
    decreases a.Length - index
  {
    m := if m>a[index] then  m else a[index];
    index := index + 1;
  }
}
```

Listing 12: Annotation Equivalence Checking Template for maxArray

```
predicate pre_original(a: array<int>,m: int)
  reads a
{
  ( a.Length >= 1)
}
```

```
predicate pre_gen(a: array<int>,m: int)
  reads a
{
  true // (#PRE) && ... (#PRE)
}

lemma pre_eq(a: array<int>,m: int)
  ensures pre_original(a,m ) <==> pre_gen(a,m )
{
}

predicate post_original(a: array<int>,m: int)
  requires pre_original(a,m)
  reads a
{
  ( forall k :: 0 <= k < a.Length ==> m >= a[k]) &&
  ( exists k :: 0 <= k < a.Length && m == a[k])
}

predicate post_gen(a: array<int>,m: int)
  requires pre_original(a,m)
  reads a
{
  true // (#POST) && ... (#POST)
}

lemma post_eq(a: array<int>,m: int)
  requires pre_original(a,m )
  requires pre_gen(a,m )
  ensures post_original(a,m ) <==> post_gen(a,m )
{
}
```

Listing 13: Filled Annotation Equivalence Checking Template for only_once

The original and generated postconditions are describing the same property: element key only appears once in the array a. But they cannot be verified equivalent by the annotation template. Lemma post_eq will fail with an empty body.

```
predicate pre_original<T(==)>(a: array<T>,key: T,b:bool)
  reads a
{
  true
}

predicate pre_gen<T(==)>(a: array<T>,key: T,b:bool)
  reads a
{
  true
}

lemma pre_eq<T(==)>(a: array<T>,key: T,b:bool)
  ensures pre_original(a,key,b ) <==> pre_gen(a,key,b )
{
}

predicate post_original<T(==)>(a: array<T>,key: T,b:bool)
  requires pre_original(a,key,b)
  reads a
{
  ( (multiset(a[..])[key] ==1 ) <==> b)
}

predicate post_gen<T(==)>(a: array<T>,key: T,b:bool)
  requires pre_original(a,key,b)
  reads a
```

```
{
  (b <==> ((exists i :: 0 <= i < a.Length && a[i] == key) && (forall i, j
     :: 0 <= i < j < a.Length && a[i] == key ==> a[j] != key)))
}

lemma post_eq<T(==)>(a: array<T>,key: T,b:bool)
  requires pre_original(a,key,b )
  requires pre_gen(a,key,b )
  ensures post_original(a,key,b ) <==> post_gen(a,key,b )
{
}
```

```

20
```

### GPT-4 System Prompt

**code2anno**:
You are an expert in Dafny.  Fill in the weakest precondition and
strongest post conditions for the dafny programs so that the dafny
programs can be verified.   Do not change provided code.  Exclude
"requires true", "requires array!=null", "requires natural number
>=0". Do not assume input array or seq is non-empty.  Do not
assume input integers are non-negative unless necessary.  Replace
the //TOFILL string with actual pre and post conditions.  Return
the whole verifiable program.

**anno2code**:
You are an expert in dafny.  You are given a dafny program with
annotations. Replace //TOFILL with the actual dafny code so that it
can be verified.  Return the whole program. If loop is needed, use
while instead of for.   DO NOT modify the function signature and
annotations.

**doc2anno**:
You are an expert in Dafny.  Generate the weakest precondition
and strongest post conditions for the dafny programs based on
the docstring.   Do not change provided code.  Exclude "requires
true", "requires array!=null", "requires natural number >=0".
Do not assume the input array or seq is non-empty.   Do not use
self-defined functions.   Do not use int.MaxValue or int.MinValue.
Do not assume input integers are non-negative unless necessary.
Return only the annotations in code format starting with ``` and
end with ```.  Do not return method implementation.

**anno2doc**:
You are an expert in dafny.   Give one docstring of the given dafny
annotation.  Make sure to capture all details described in the
annotation.

**code2doc**:
You are an expert in dafny.   Give one docstring of the given dafny
code's functional behavior.  Do not mention implementation details.
Assume 'assert' as preconditions.

**doc2code**:
You are an expert in dafny.  You are given a dafny program
docstring. If loop is needed, use while instead of for. Replace
//TOFILL with the actual dafny code without annotation.  Return the
whole program.

**docstring equivalence checker**:
Determine if two docstrings describe the exact same functional
behavior.   Make sure all details are exactly
the same.

## A.6 C1 SUB-CATEGORIES

| C1 | pre- too weak | pre- good | pre- too strong | pre- wrong |
|---|---|---|---|---|
| post- too weak | uncertain | verify | verify | fail |
| post- good | fail | - | verify | fail |
| post- too strong | fail | fail | uncertain | fail |
| post- wrong | fail | fail | fail | fail |

Table 6: C1 sub-categories.

## A.7 ADDITIONAL EXPERIMENT RESULTS

In Table 7, we show all six checks results for each example in our dataset CloverBench. Table 10 shows variant C1 are all rejected. Table 11 shows variant C2 are all rejected. Table 12 shows variant C4 are all rejected. Table 8 shows code generation ablation studies. Table 9 shows annotation generation ablation studies.

| Row No. | test | anno-complete | doc2anno | doc2code | anno2doc | code2doc | 3-edges |
|---|---|---|---|---|---|---|---|
| 1 | abs | A | A | A | A | A | A |
| 2 | all_digits | A | A | A | A | A | A |
| 3 | array_append | A | A | A | A | A | A |
| 4 | array_concat | A | A | A | A | A | A |
| 5 | array_copy | A | A | A | A | A | A |
| 6 | array_product | A | A | A | A | A | A |
| 7 | array_sum | A | A | A | A | A | A |
| 8 | avg | A | A | A | A | A | A |
| 9 | below_zero | A | A | A | A | A | A |
| 10 | binary_search | A | A | A | A | A | A |
| 11 | bubble_sort | A | A | A | A | A | A |
| 12 | cal_ans | A | A | A | A | A | A |
| 13 | cal_sum | A | A | A | A | A | A |
| 14 | canyon_search | A | A | R | A | A | R |
| 15 | compare | A | A | A | A | A | A |
| 16 | convert_map_key | R | R | R | A | A | R |
| 17 | copy_part | A | A | A | A | A | A |
| 18 | count_lessthan | R | R | R | A | A | R |
| 19 | double_quadruple | A | A | R | A | A | R |
| 20 | even_list | R | R | A | A | A | R |
| 21 | find | A | A | A | A | A | A |
| 22 | has_close_elements | A | A | A | A | A | A |
| 23 | insert | A | A | A | A | A | A |
| 24 | integer_square_root | A | A | A | A | A | A |
| 25 | is_even | A | A | A | A | A | A |
| 26 | is_palindrome | A | A | A | A | A | A |
| 27 | linear_search1 | A | A | A | A | A | A |
| 28 | linear_search2 | A | A | A | A | A | A |
| 29 | linear_search3 | A | A | A | A | A | A |
| 30 | longest_prefix | A | A | A | A | A | A |
| 31 | max_array | A | A | A | A | A | A |
| 32 | min_array | A | A | A | A | A | A |
| 33 | min_of_two | A | A | A | A | A | A |
| 34 | modify_2d_array | A | R | A | A | R | R |
| 35 | multi_return | A | A | R | A | A | R |
| 36 | online_max | R | R | R | A | A | R |
| 37 | only_once | A | R | A | A | A | R |
| 38 | quotient | A | A | A | A | A | A |
| 39 | remove_front | A | A | A | A | A | A |
| 40 | replace | A | R | A | A | A | R |
| 41 | return_seven | A | A | A | A | A | A |
| 42 | reverse | A | A | A | A | A | A |
| 43 | rotate | A | A | A | A | A | A |
| 44 | selectionsort | A | A | A | A | A | A |
| 45 | seq_to_array | R | A | R | A | A | R |
| 46 | set_to_seq | R | A | R | A | A | R |
| 47 | slope_search | A | R | R | A | R | R |
| 48 | swap | A | A | R | A | A | R |
| 49 | swap_arith | A | A | A | A | A | A |
| 50 | swap_bitvector | A | A | A | A | A | A |
| 51 | swap_in_array | A | A | A | A | A | A |
| 52 | swap_sim | A | A | A | A | A | A |
| 53 | test_array | A | A | A | A | A | A |
| 54 | triple | A | A | A | A | A | A |
| 55 | triple2 | A | A | A | A | A | A |
| 56 | triple3 | A | A | A | A | A | A |
| 57 | triple4 | A | A | A | A | A | A |
| 58 | two_sum | A | A | A | A | A | A |
| 59 | update_array | A | A | A | A | A | A |
| 60 | update_map | R | R | R | A | A | R |

Table 7: COLVER ground truth experiments: Each row represents one example. Each column represents one directed edge. In each cell, there is either **A** or **R**. **A** means that the directed edge is accepted by our checker and **R** means that the cell is rejected by our checker. In each row, if all edges are accepted, then the example is accepted.

| Row No. | test | max3tries | oneTry | noVerify_max3tries | withSpec_max3tries |
|---|---|---|---|---|---|
| 1 | abs | A | A | A | A |
| 2 | all_digits | A | A | A | A |
| 3 | array_append | A | A | A | A |
| 4 | array_concat | A | A | A | A |
| 5 | array_copy | A | A | A | A |
| 6 | array_product | A | R | A | A |
| 7 | array_sum | A | R | A | A |
| 8 | avg | A | A | R | A |
| 9 | below_zero | A | R | R | A |
| 10 | binary_search | A | A | A | A |
| 11 | bubble_sort | A | A | A | A |
| 12 | cal_ans | A | R | A | A |
| 13 | cal_sum | A | A | A | A |
| 14 | canyon_search | A | R | R | A |
| 15 | compare | A | A | A | A |
| 16 | convert_map_key | R | R | R | R |
| 17 | copy_part | A | A | A | A |
| 18 | count_lessthan | R | R | R | R |
| 19 | double_quadruple | A | A | A | A |
| 20 | even_list | R | R | R | R |
| 21 | find | A | A | A | A |
| 22 | has_close_elements | A | R | R | A |
| 23 | insert | A | R | A | A |
| 24 | integer_square_root | A | R | A | A |
| 25 | is_even | A | A | A | A |
| 26 | is_palindrome | A | A | A | A |
| 27 | linear_search1 | A | A | A | A |
| 28 | linear_search2 | A | A | A | A |
| 29 | linear_search3 | A | A | A | A |
| 30 | longest_prefix | A | A | A | A |
| 31 | max_array | A | A | A | A |
| 32 | min_array | A | A | A | A |
| 33 | min_of_two | A | A | A | A |
| 34 | modify_2d_array | A | A | A | A |
| 35 | multi_return | A | A | A | A |
| 36 | online_max | R | R | R | R |
| 37 | only_once | A | A | A | A |
| 38 | quotient | A | A | A | A |
| 39 | remove_front | A | A | A | A |
| 40 | replace | A | A | A | A |
| 41 | return_seven | A | A | A | A |
| 42 | reverse | A | R | A | A |
| 43 | rotate | A | A | A | A |
| 44 | selectionsort | A | A | A | A |
| 45 | seq_to_array | R | R | R | R |
| 46 | set_to_seq | R | R | R | R |
| 47 | slope_search | A | A | A | A |
| 48 | swap | A | A | A | A |
| 49 | swap_arith | A | R | A | A |
| 50 | swap_bitvector | A | R | A | A |
| 51 | swap_in_array | A | A | A | A |
| 52 | swap_sim | A | A | A | A |
| 53 | test_array | A | A | A | A |
| 54 | triple | A | A | A | A |
| 55 | triple2 | A | A | A | A |
| 56 | triple3 | A | A | A | A |
| 57 | triple4 | A | A | A | A |
| 58 | two_sum | A | R | R | A |
| 59 | update_array | A | A | A | A |
| 60 | update_map | R | R | R | R |

Table 8: Ablation studies that compare code generation under different configurations. Each column represents one configuration. We have: a maximum of 3 tries with verifier feedback, the first try, a maximum of 3 tries with only compiler and no verifier feedback, and a maximum of 3 tries with verifier feedback plus docstrings.

| Row No. | test | max3tries | oneTry |
|---|---|---|---|
| 1 | abs | A | A |
| 2 | all_digits | A | A |
| 3 | array_append | A | A |
| 4 | array_concat | A | A |
| 5 | array_copy | A | A |
| 6 | array_product | A | A |
| 7 | array_sum | A | R |
| 8 | avg | A | A |
| 9 | below_zero | R | R |
| 10 | binary_search | A | R |
| 11 | bubble_sort | R | R |
| 12 | cal_ans | A | A |
| 13 | cal_sum | A | R |
| 14 | canyon_search | A | R |
| 15 | compare | A | R |
| 16 | convert_map_key | R | R |
| 17 | copy_part | R | R |
| 18 | count_lessthan | R | R |
| 19 | double_quadruple | A | R |
| 20 | even_list | R | R |
| 21 | find | A | A |
| 22 | has_close_elements | A | A |
| 23 | insert | R | R |
| 24 | integer_square_root | A | R |
| 25 | is_even | A | A |
| 26 | is_palindrome | R | R |
| 27 | linear_search1 | A | A |
| 28 | linear_search2 | A | R |
| 29 | linear_search3 | R | R |
| 30 | longest_prefix | R | R |
| 31 | max_array | A | R |
| 32 | min_array | A | A |
| 33 | min_of_two | A | A |
| 34 | modify_2d_array | R | R |
| 35 | multi_return | A | A |
| 36 | online_max | R | R |
| 37 | only_once | R | R |
| 38 | quotient | A | A |
| 39 | remove_front | A | R |
| 40 | replace | R | R |
| 41 | return_seven | A | A |
| 42 | reverse | R | R |
| 43 | rotate | A | R |
| 44 | selectionsort | R | R |
| 45 | seq_to_array | A | A |
| 46 | set_to_seq | R | R |
| 47 | slope_search | R | R |
| 48 | swap | A | A |
| 49 | swap_arith | A | R |
| 50 | swap_bitvector | A | A |
| 51 | swap_in_array | A | A |
| 52 | swap_sim | A | A |
| 53 | test_array | A | A |
| 54 | triple | A | A |
| 55 | triple2 | A | A |
| 56 | triple3 | A | A |
| 57 | triple4 | A | A |
| 58 | two_sum | A | R |
| 59 | update_array | A | A |
| 60 | update_map | R | R |

Table 9: Ablation studies for annotation generation from pure code: Note that this is the only place where we count loop invariants in annotations. **A** means (1) generated annotations are equivalent to the original and (2) generated loop invariants are enough to have the code verified by Dafny.

| Row No. | test | anno-sound | anno-complete | doc2anno | doc2code | anno2doc | code2doc |
|---|---|---|---|---|---|---|---|
| 1 | abs | A | R | R | A | R | A |
| 2 | all_digits | R | R | R | A | R | A |
| 3 | array_append | A | A | R | A | A | A |
| 4 | array_concat | A | A | R | A | A | A |
| 5 | array_copy | A | A | R | A | A | A |
| 6 | array_product | R | R | R | A | A | A |
| 7 | array_sum | R | A | R | A | A | A |
| 8 | avg | R | A | R | A | R | A |
| 9 | below_zero | A | R | R | A | R | A |
| 10 | binary_search | R | A | R | A | A | A |
| 11 | bubble_sort | R | A | R | A | R | A |
| 12 | cal_ans | A | R | R | A | R | A |
| 13 | cal_sum | A | R | R | A | R | A |
| 14 | canyon_search | R | R | R | R | R | A |
| 15 | compare | A | A | R | A | A | A |
| 16 | convert_map_key | R | R | R | R | R | A |
| 17 | copy_part | A | A | R | A | R | A |
| 18 | count_lessthan | A | R | R | R | A | A |
| 19 | double_quadruple | A | A | R | R | R | A |
| 20 | even_list | R | R | R | A | R | A |
| 21 | find | A | A | R | A | A | A |
| 22 | has_close_elements | A | A | R | A | A | A |
| 23 | insert | A | A | R | A | R | A |
| 24 | integer_square_root | R | A | R | A | A | A |
| 25 | is_even | R | R | R | A | R | A |
| 26 | is_palindrome | A | A | R | A | A | A |
| 27 | linear_search1 | A | A | R | A | A | A |
| 28 | linear_search2 | R | A | R | A | A | A |
| 29 | linear_search3 | R | A | R | A | R | A |
| 30 | longest_prefix | R | A | R | A | A | A |
| 31 | max_array | R | R | R | A | R | A |
| 32 | min_array | R | A | R | A | A | A |
| 33 | min_of_two | A | A | R | A | A | A |
| 34 | modify_2d_array | R | A | R | A | A | R |
| 35 | multi_return | A | A | R | R | R | A |
| 36 | online_max | R | R | R | R | R | A |
| 37 | only_once | R | A | R | A | A | A |
| 38 | quotient | R | A | R | A | A | A |
| 39 | remove_front | R | A | R | A | A | A |
| 40 | replace | R | R | R | A | R | A |
| 41 | return_seven | A | A | R | A | A | A |
| 42 | reverse | R | A | R | A | R | A |
| 43 | rotate | A | A | R | A | A | A |
| 44 | selectionsort | R | A | R | A | A | A |
| 45 | seq_to_array | A | R | R | R | A | A |
| 46 | set_to_seq | R | R | R | R | R | A |
| 47 | slope_search | A | A | R | R | A | R |
| 48 | swap | A | A | R | R | R | A |
| 49 | swap_arith | R | R | R | A | A | A |
| 50 | swap_bitvector | A | R | R | A | R | A |
| 51 | swap_in_array | R | R | R | A | R | A |
| 52 | swap_sim | R | R | R | A | R | A |
| 53 | test_array | R | R | R | A | R | A |
| 54 | triple | R | R | R | A | R | A |
| 55 | triple2 | A | A | R | A | R | A |
| 56 | triple3 | A | A | R | A | R | A |
| 57 | triple4 | A | A | R | A | A | A |
| 58 | two_sum | A | R | R | A | R | A |
| 59 | update_array | R | R | R | A | R | A |
| 60 | update_map | R | R | R | R | A | A |

Table 10: C1: If any one of the cells in a row is rejected, then the inconsistent example is successfully rejected. In this table, all examples are rejected.

| Row No. | test | anno-sound | anno-complete | doc2anno | doc2code | anno2doc | code2doc |
|---|---|---|---|---|---|---|---|
| 1 | abs | A | A | R | A | R | R |
| 2 | all_digits | A | A | R | R | R | R |
| 3 | array_append | A | A | R | A | R | R |
| 4 | array_concat | A | A | R | A | A | R |
| 5 | array_copy | A | A | R | A | R | A |
| 6 | array_product | A | A | R | A | A | R |
| 7 | array_sum | A | A | R | A | A | A |
| 8 | avg | A | A | R | R | R | R |
| 9 | below_zero | A | A | R | R | R | R |
| 10 | binary_search | A | A | R | R | A | R |
| 11 | bubble_sort | A | A | R | A | R | R |
| 12 | cal_ans | A | A | R | R | R | R |
| 13 | cal_sum | A | A | R | R | R | R |
| 14 | canyon_search | A | R | R | R | R | R |
| 15 | compare | A | A | R | A | R | R |
| 16 | convert_map_key | A | R | R | R | A | A |
| 17 | copy_part | A | A | R | R | R | R |
| 18 | count_lessthan | A | R | R | R | R | R |
| 19 | double_quadruple | A | A | R | A | R | R |
| 20 | even_list | A | R | R | A | A | A |
| 21 | find | A | A | R | A | R | R |
| 22 | has_close_elements | A | A | R | R | R | R |
| 23 | insert | A | A | R | R | R | R |
| 24 | integer_square_root | A | A | R | R | R | A |
| 25 | is_even | A | A | R | A | A | R |
| 26 | is_palindrome | A | A | R | R | R | R |
| 27 | linear_search1 | A | A | R | R | A | R |
| 28 | linear_search2 | A | A | R | A | R | R |
| 29 | linear_search3 | A | A | R | A | A | R |
| 30 | longest_prefix | A | A | R | A | A | A |
| 31 | max_array | A | A | R | A | R | R |
| 32 | min_array | A | A | R | R | R | R |
| 33 | min_of_two | A | A | R | R | R | R |
| 34 | modify_2d_array | A | A | R | R | R | R |
| 35 | multi_return | A | A | R | R | R | R |
| 36 | online_max | A | R | R | R | R | R |
| 37 | only_once | A | A | R | R | R | R |
| 38 | quotient | A | A | R | A | A | A |
| 39 | remove_front | A | A | R | R | R | R |
| 40 | replace | A | A | R | R | R | R |
| 41 | return_seven | A | A | R | R | R | R |
| 42 | reverse | A | A | R | R | R | R |
| 43 | rotate | A | A | R | R | R | A |
| 44 | selectionsort | A | A | R | A | A | R |
| 45 | seq_to_array | A | R | R | R | R | R |
| 46 | set_to_seq | A | R | R | A | R | R |
| 47 | slope_search | A | A | R | R | R | R |
| 48 | swap | A | A | A | A | R | R |
| 49 | swap_arith | A | A | A | R | R | R |
| 50 | swap_bitvector | A | A | R | R | R | R |
| 51 | swap_in_array | A | A | R | R | R | R |
| 52 | swap_sim | A | A | R | A | R | R |
| 53 | test_array | A | A | R | R | R | R |
| 54 | triple | A | A | R | R | R | R |
| 55 | triple2 | A | A | R | A | A | R |
| 56 | triple3 | A | A | A | A | R | R |
| 57 | triple4 | A | A | R | R | R | R |
| 58 | two_sum | A | A | R | R | R | R |
| 59 | update_array | A | A | R | R | R | R |
| 60 | update_map | A | R | R | A | R | R |

Table 11: C2: all examples are rejected

| Row No. | test | anno-sound | anno-complete | doc2anno | doc2code | anno2doc | code2doc |
|---|---|---|---|---|---|---|---|
| 1 | abs | A | A | R | R | R | R |
| 2 | all_digits | A | R | R | A | A | A |
| 3 | array_append | A | R | R | R | A | A |
| 4 | array_concat | A | R | R | R | R | R |
| 5 | array_copy | A | R | R | R | A | R |
| 6 | array_product | A | A | R | R | R | R |
| 7 | array_sum | A | R | R | R | R | R |
| 8 | avg | A | R | R | R | R | R |
| 9 | below_zero | A | R | R | R | R | R |
| 10 | binary_search | A | R | R | R | R | R |
| 11 | bubble_sort | A | A | R | R | R | R |
| 12 | cal_ans | A | R | R | R | R | A |
| 13 | cal_sum | A | A | R | R | R | R |
| 14 | canyon_search | A | R | R | R | A | A |
| 15 | compare | A | R | R | R | R | R |
| 16 | convert_map_key | A | R | R | R | A | R |
| 17 | copy_part | A | R | R | R | R | R |
| 18 | count_lessthan | A | R | R | R | R | A |
| 19 | double_quadruple | A | R | R | R | R | R |
| 20 | even_list | A | R | R | R | R | R |
| 21 | find | A | A | R | A | A | R |
| 22 | has_close_elements | A | R | R | A | A | A |
| 23 | insert | A | R | R | R | R | R |
| 24 | integer_square_root | A | R | R | R | R | R |
| 25 | is_even | A | R | R | A | A | A |
| 26 | is_palindrome | A | R | R | A | A | A |
| 27 | linear_search1 | A | R | R | R | R | R |
| 28 | linear_search2 | A | R | R | R | A | R |
| 29 | linear_search3 | A | A | R | R | R | R |
| 30 | longest_prefix | A | R | R | R | R | R |
| 31 | max_array | A | R | R | R | R | R |
| 32 | min_array | A | A | R | A | A | A |
| 33 | min_of_two | A | A | R | R | R | A |
| 34 | modify_2d_array | A | R | R | R | R | R |
| 35 | multi_return | A | A | R | R | R | R |
| 36 | online_max | A | R | R | R | R | R |
| 37 | only_once | A | A | R | R | R | A |
| 38 | quotient | A | R | R | R | A | R |
| 39 | remove_front | A | R | R | R | R | A |
| 40 | replace | A | R | R | R | A | R |
| 41 | return_seven | A | A | R | R | R | R |
| 42 | reverse | A | R | R | A | A | A |
| 43 | rotate | A | A | R | R | A | R |
| 44 | selectionsort | A | R | R | R | R | R |
| 45 | seq_to_array | A | R | R | R | R | A |
| 46 | set_to_seq | A | R | R | R | R | A |
| 47 | slope_search | A | R | R | R | R | R |
| 48 | swap | A | R | R | R | R | A |
| 49 | swap_arith | A | R | R | R | R | R |
| 50 | swap_bitvector | A | A | R | R | R | R |
| 51 | swap_in_array | A | R | R | R | A | R |
| 52 | swap_sim | A | R | R | R | R | R |
| 53 | test_array | A | R | R | R | R | R |
| 54 | triple | A | R | R | R | R | R |
| 55 | triple2 | A | R | R | R | R | R |
| 56 | triple3 | A | R | R | R | R | R |
| 57 | triple4 | A | R | R | R | R | R |
| 58 | two_sum | A | R | R | A | A | A |
| 59 | update_array | A | R | R | R | R | R |
| 60 | update_map | A | R | R | R | A | R |

Table 12: C4: all examples are rejected

