# OpenReview forum: "Clover: Closed-Loop Verifiable Code Generation"
_ICLR.cc/2024/Conference — ICLR 2024 Conference Withdrawn Submission_

### Official Review · Reviewer_L24Y · 2023-10-31

**Soundness:** 2 fair
**Presentation:** 4 excellent
**Contribution:** 3 good
**Rating:** 5
**Confidence:** 5

**Summary:**

The paper presents a technique for "verifiable code generation" in which rather than the generated artifact being code alone, one should think about an artifact that consists of a triplet (code, docstring, annotation).

The core idea is to use a consistency checker to verify that the triplet (c,d,a) is consistent. To do that, Clover performs a 6-way check:
1. code → docstring
2. docstring → code
3. code → annotation
4. annotation → code
5. docstring → annotation
6. annotation → docstring

The paper also presents the CloverBench, a set of 60 small textbook-style programs (e.g., binary_search, bubble_sort).

**Strengths:**

- An interesting approach for making generated code carry supporting evidence for its correctness.
- A benchmark suite that can enable further experimentation on (code,docstring,annotation) triplets or subsets thereof.
- Clear and compelling presentation.

**Weaknesses:**

- I like the approach, but I'm afraid that the evaluation really falls short. The main weakness is that CloverBench contains classical examples for which GPT4 is clearly able to generate docstring, invariants, and code quite easily from each component of the triplet. In fact, I am guessing that it has seen different versions of the loop-invariant for each of these examples multiple times in different styles and languages.
- The weakest link in the Clover approach itself is the natural language description in the docString. Would realistic docStrings contain the level of details required for correspondence with the annotation? What happens when they do not? The under-specification gap between a docString and the annotation may be hard to bridge?
- Equivalence checking is always a challenge. This challenge is even more pronounced for checking "equivalence" between natural language descriptions.
- The Dafny syntax gets in the way of getting a clear reading of where are the problems. Given the simple examples, I would hypothesize that the Clover would have seen 100% success with GPT4 and a mainstream language for annotations.

**Questions:**

- Your docStrings (e.g., as shown in Listing 2) are quite elaborate and are much closer to the formal spec than to a natural language docString that you'd expect to find in real programs. The distance between realistic NL docStrings and the formal annotation can be a real challenge for Clover. Do you have any thoughts on how to address that?
- Maybe you can consider Clover with a set of input/output examples instead of or in addition to the docString. Maybe it makes sense to consider (code,annotation, docString+examples) either as a triplet in which examples make the docString more robust, or as a quadruple in which examples provide yet another aspect, making it a four-leaf clover.
- When all three artifacts (code,docstring, annotation) are generated by the same source, they may be consistent but incorrect. Did you consider using a combination of different models for generating the triplet?
- Why don't you repeat the experiments with an annotation language embedded into a mainstream language as a library / helper functions? Something that would be then easy to translate back to Dafny. Alternatively, use the prompt to "teach" GPT4 some basic Dafny syntax and its meaning.

---

> ### Author Response · Authors · 2023-11-16
>
> > I like the approach, but I'm afraid that the evaluation really falls short. The main weakness is that CloverBench contains classical examples for which GPT4 is clearly able to generate docstring, invariants, and code quite easily from each component of the triplet. In fact, I am guessing that it has seen different versions of the loop-invariant for each of these examples multiple times in different styles and languages.
>
> LLMs generalize based on examples they have seen.  In order for Clover to work, the LLMs have to be able to understand the code, annotations, and docstrings.  Since this is a new direction, it is unclear how far LLMs can generalize.  We argue that our evaluation is the right first step.  Note that Dafny is not a mainstream language, and all of our examples were hand-written.  This means that the LLM has never seen these specific examples.  The fact that it works at all answers a first set of questions about whether LLMs can reason well enough to add specifications in a new language to code that they have never seen (but which may be similar to code that they have seen). The fact that LLMs can do this is a new result and is not at all obvious.  Clearly, the next step will be to scale up to more difficult examples, but we believe it would be a mistake to jump directly to complicated code without tackling smaller milestones first.
>
> As for the use of other languages, please refer to our general response to Question 3.
>
> > The weakest link in the Clover approach itself is the natural language description in the docString. Would realistic docStrings contain the level of details required for correspondence with the annotation? What happens when they do not? The under-specification gap between a docString and the annotation may be hard to bridge?
>
> This is a great question that we did encounter when we conducted the experiments. The docstring in our context is not simply informal natural language.  It must be a bit more than that: specifically it should be  a complete natural language description of the functional properties of the code. The annotation is treated as a formal certificate, and the natural language description bridges the formal properties and human intention. We will make this point clearer in the revised version.
>
>
> > Equivalence checking is always a challenge. This challenge is even more pronounced for checking "equivalence" between natural language descriptions.
>
> For our thoughts on docstring equivalence check, please refer to our general response in Question 2.
>
> This is true, and especially the “equivalence” between natural language descriptions is imprecise, as we also mention in the paper. The result from GPT4 is skewed towards accept, which means it is easy to say a pair of docstrings are equivalent if there is no contradiction. Therefore, the rejection rate for incorrect examples on the edges that rely on docstring equivalence checks is not ideal. However, the equivalence check for annotations is strict, and the equivalence check for code is okay, so the aggregated rejection rate still looks good (i.e., 100% in our experiments).
>
>
> > The Dafny syntax gets in the way of getting a clear reading of where are the problems. Given the simple examples, I would hypothesize that the Clover would have seen 100% success with GPT4 and a mainstream language for annotations.
>
> Please refer to our general response, Question 5. We do observe Dafny syntax errors as a significant cause of errors, but we also observe cases where the model failed to understand the program correctly. Moreover, since we allow feedback from the Dafny compiler to GPT4, more than half of the time, GPT4 is able to fix the syntax mistakes if it has the correct understanding of the program functionality.  So Dafny syntax is part of the problem, but not all of it.  We had to balance the ability of GPT4 to reason about Dafny with the need for a strong verification back-end.  Dafny was a good compromise.  If we had used a mainstream language, there are several other failure modes we would likely have seen.  First of all, the annotation language may not be strong enough (e.g., Dafny supports quantifiers, but C-style assertions do not).  Second, the verification tools may have failed.  Dafny is SoTA for deductive verification.  Tools for other languages fall short in this respect.

---

> ### Author Response · Authors · 2023-11-16
>
> > Your docStrings (e.g., as shown in Listing 2) are quite elaborate and are much closer to the formal spec than to a natural language docString that you'd expect to find in real programs. The distance between realistic NL docStrings and the formal annotation can be a real challenge for Clover. Do you have any thoughts on how to address that?
>
> In our hypothetical setting of an ideal codebase, the docstring is interpreted as a natural language description of the formal specification (annotation). We do expect that the docstring must contain all the information needed for a human developer to recreate the correct code. We’ll elaborate on this in a revised version.   Note that one interesting direction for future work would be to explore whether LLMs can bridge this gap between informal docstrings and the more formal ones we require.
>
> > Maybe you can consider Clover with a set of input/output examples instead of or in addition to the docString. Maybe it makes sense to consider (code,annotation, docString+examples) either as a triplet in which examples make the docString more robust, or as a quadruple in which examples provide yet another aspect, making it a four-leaf clover.
>
> Thanks for the great suggestion! In Appendix A.2.2 (page 15), we have briefly discussed the option of adding unit tests in the loop (four-leaf clover!). In the case of the user entering the docstring, it makes a lot of sense to add input/output examples, which can hugely improve the accuracy of the docstring, which is also common in other datasets for code (MBPP, HumanEval). In the case of machine-generated input/output examples, it may not be easier than generating the annotations. We will seriously consider this as a next step.
>
> > When all three artifacts (code,docstring, annotation) are generated by the same source, they may be consistent but incorrect. Did you consider using a combination of different models for generating the triplet?
>
> This is a good point. Using a portfolio of models is a great suggestion for reducing consistent model hallucinations We have also experimented with Claude. Our hypothesis is that an incorrect triplet is hard to get consistent among each artifact accidentally, from the intuition that the possibility of being incorrect is many. But theoretically, such a case could happen, which we also discussed in Appendix A.2.1 (page 15). The reduction from correctness to consistency is not mathematically complete, which is an inherent limitation of the Clover framework. A combination of different models may help with the problem, though it further complicates the process. Also, as we discussed in Appendix A.2.2, there could be many Clover variants using the principle of consistency test. Ensembling different models could be an interesting future research.
>
> > Why don't you repeat the experiments with an annotation language embedded into a mainstream language as a library / helper functions? Something that would be then easy to translate back to Dafny. Alternatively, use the prompt to "teach" GPT4 some basic Dafny syntax and its meaning.
>
> Please refer to our general response to Question 5. We have also tried to “teach” GPT4 some basic Dafny syntax as we showed in the system prompt but they tend to not work effectively. However, given our evidence in Question 5, we think that Dafny syntax is not a significant issue given our current use.
>
> It is a great suggestion to translate a mainstream annotation language to Dafny. But there does not exist an automatic translator. If we were to use LLM for translation, the outcome would not be better than our current use. We will provide experimental evidence to show that translation is not that easy. We tried to translate the 18 typical loop invariants in table 2 (Pei 2023) and only 8 succeded out of 18 in the first try and 13 succeeded after three tries with Dafny compiler feedback. This shows that automatic translation using LLM to Dafny syntax is not easier than directly generating invariants in Dafny syntax.
>
>
> Kexin Pei, David Bieber, Kensen Shi, Charles Sutton, and Pengcheng Yin. 2023. Can large language models reason about program invariants? In Proceedings of the 40th International Conference on Machine Learning (ICML'23), Vol. 202. JMLR.org, Article 1144, 27496–27520.

---

> ### Author Response · Authors · 2023-11-21
>
> Dear reviewer,
>
> We are really encouraged by your helpful and positive feedback and sincerely appreciate your efforts. We hope our answer, updated experiments, and the corresponding manuscript make the paper clearer and stronger. Please let us know if you have any further comments, and we are more than happy to address them.
>
> Best,
>
> Submission113 authors

---

### Official Review · Reviewer_29hf · 2023-11-01

**Soundness:** 3 good
**Presentation:** 3 good
**Contribution:** 2 fair
**Rating:** 5
**Confidence:** 4

**Summary:**

Summary:
-------------
This paper introduces the Clover paradigm, which aims to ensure the correctness of code generated by large language models (LLMs), through a closed-loop verification process. As the process of formal verification of code is tedious when done manually by human experts, the authors propose an LLM-based automated approach for the same. The paper provides an interesting insight that AI-generated codes (i.e. C) should inherently also include formal specifications (annotation i.e. A which includes precondition, postcondition, loop-invariants) and natural language descriptions (docstring i.e. D). As such, the six ordered pairs of these three components (code, annotation, docstring) viz. A-D, D-A, C-D, D-C, A-C, and C-A can be mutually verified for consistency through either a LLM or a SAT/SMT solver. Therefore, a code can be considered formally verified when all these six consistency checks are passed. Essentially, the Clover checks act as a conservative filter that approves of codes that are formally verified, well-documented, and internally consistent.

The paper also puts forward a miniature CloverBench dataset of 60 codes written in the Dafny programming language. The codes are small-sized and contain at most one method or helper function (termed as those 'found in standard CS textbooks'). Each code has a ground-truth version (where the code, annotation, and docstring are manually verified to be correct and consistent), three incorrect versions, and five unit tests.

An input program is assumed to contain annotation (A), documentation (D), and code (C). As part of the Clover paradigm, the paper proposes how each of the six ordered pairs among A, D, and C can be mutually verified. For instance, the ordered pair A-D can be verified using an LLM. Given an annotation (A), an LLM is asked to generate a docstring (D_gen). Subsequently, it is checked whether D and D_gen are equivalent. Such LLM-based generation and equivalence checking is carried out for A-D, D-A, C-D, D-C, and A-C ordered pairs. However, for the C-A ordered pair, Dafny's deductive verifier is used to check whether the code satisfies the generated annotation. The paper also elaborates on how equivalence can be tested between each input component and its corresponding generated version viz. (C, C_gen), (A, A_gen), and (D, D_gen). (C, C_gen) equivalence is tested using the unit tests included as part of the Clover dataset. (A, A_gen) equivalence is tested by writing the equivalence of the two annotations as a formal lemma and asking Dafny's formal tool to prove the lemma. (D, D_gen) equivalence is tested by asking an LLM whether the two docstrings are semantically equivalent. The LLM used for all these checks is GPT-4.

For evaluation, the authors first check whether GPT-4 can generate code with annotations and docstrings included. Using a zero-shot approach, given the annotation the LLM can produce the correct code for 41/60 in the dataset. This accuracy iteratively increases when the LLM is provided with the Dafny compiler and verifier's output as feedback and the LLM is repeatedly asked to correct itself. A similar pattern is observed when the LLM is asked to generate an annotation given a Dafny code. Secondly, the author evaluates their Clover paradigm using the six consistency checks. The Clover paradigm accepts 45 of the 60 correct codes in the CloverBench dataset and rejects all the incorrect versions. The major reason for non-acceptance is incorrect syntax in the LLM-generated Dafny codes. The paper also puts forward a fairly elaborate ablation study on the different aspects of the Clover paradigm.

The paper also discusses the limitations of the current implementation of the Clover paradigm, including the need for more research to improve the formal verification tools and to expand the coverage to more mainstream languages. However, the authors argue that the Clover paradigm has the potential to significantly reduce the time and expertise required for formal verification and to improve the trustworthiness of code generated by LLMs.

**Strengths:**

- The paper is very nicely written and proposes a well-principled approach to verify AI-generated codes.

- Particularly, I found the insight interesting where the authors propose that AI-generated codes should inherently also include formal specifications (annotation viz. precondition, postcondition, loop-invariants) and natural language descriptions (docstring). This is a principled strategy that essentially asks an LLM to provide some form of certification justifying what it generated.

- The proposed set of six checks in the Clover paradigm has the potential towards a good foundation for verifying AI-generated codes, especially because LLM-generated codes are typically erroneous when the complexity of codes increases.

**Weaknesses:**

- Although the paper proposes a good concept for program verification, it's still in a nascent stage and has a long way to go for real-world deployment. I understand that this is a first step towards a bigger objective when the author mentions that "we predict that Clover, coupled with steadily improving generative AI and formal tools, will enable a future in which fully automatic, scalable generation of formally verified code is feasible. This paper charts the first steps towards realizing this vision". However, a few aspects of the paper concern me. For example, the Clover paradigm is tested on a simple and very small dataset of 60 programs that contain at most one helper function (or even none). This is a significantly trivial benchmark to test on, even for a proof of concept. In my opinion, for the readers to be convinced of the applicability of the Clover paradigm, it should be tested on relatively larger benchmarks.

- Even for a proof of concept, a paper should argue how it can suffice for challenging cases. When a paper aims to propose a promising program verification approach, it is not sufficient to say that "This test is, of course, imprecise (complete but not sound), but our evaluation suggests that it suffices for the level of complexity in CloverBench. More advanced equivalence checking techniques might be required for more complex examples." or "This method is conservative in the sense that it succeeds only if the two annotations are indeed equivalent, but it may fail on equivalent annotations due to limitations of the verification tool being used.". It would be nice to get an insight into how the authors aim to check (or at least approximately) code equivalence for difficult benchmarks.

- The paper provides no insight into the performance of Clover on other mainstream programming languages like C, C++, Java, and Python. Although I agree with the fact that AI-generated codes should include formal specifications and natural language descriptions which is inherent in a language like Dafny that supports formal specification, there should be at least some insight about how this should be tackled in or extended to the mainstream languages of today. Otherwise, this would essentially mean that to get the benefit of a closed-loop verifier for AI-generated codes, the whole community should shift towards a verifier-friendly language like Dafny, which is not possible.

- The related work of the paper is not elaborate enough and fails to describe in which line of work the present paper is closest to. Is the paper the first to propose a verifier for (AI/LLM)-based code generation from docstrings and annotations? If not, what were the limitations in the other papers that motivated the authors to come up with this approach? The related work will also look better if structured by organizing it into comprehensive categories that encompass various code verification methods in use.

- The paper does not compare their proposed approach against other similar works in program verification. The input requirements for Clover are quite strong (in the sense that it requires annotation and documentation along with the code). So, an AI-generated program that has all these three can be verified by Clover and any other similar SoTA program verification approaches that do not have such strong requirements. As such, it makes sense to provide a comprehensive comparison with SoTA approaches and justify empirically whether there is a significant improvement in verification filtering after imposing the strong input requirements. If not, why will the strong input requirements make sense for an automated verifier?

- Although the paper is well-written, I believe the structure could be made more reader-friendly. It needed two passes for me to understand the concept. The insights given in the later part of the introduction felt too abstract on the initial read. It required me to go through the whole proposed approach and come back to the introduction again to grasp the insights given in the introduction. Algorithm-1 is helpful in this context, but I presume that was pushed to the appendix due to lack of space.

- Consider a situation: A user provides a prompt to the LLM to generate a code according to some specification (e.g. sort the input array in ascending order). The LLM generates a code that sorts the input array in descending order. Along with the code, the LLM provides annotation and docstring that are fully consistent with the generated code. In this case, the Clover checks will be successful because the code-annotation-docstring is mutually consistent with the six checks. But, the code does not match the user specifications. Because the paper proposes a closed-loop approach of code generation + verification, it would be insightful to tackle such odd but common cases where the generated annotation, docstring, and code are mutually consistent, but do not tally with the prompt given by the user to generate the code (original functionality intended by the user).

- (Minor) There are a few spelling and syntactic mistakes in the current version e.g. "tranlations" on Page 6, "COLVER" on Page 22, and space missing before full-stops in quite a few places.

- Reproducibility: I did not see any URL to access the code or dataset.

**Questions:**

Is the paper the first to propose a verifier for (AI/LLM)-based code generation from docstrings and annotations? If not, what were the limitations in the other papers that motivated the authors to come up with this approach?

---

> ### Author Response · Authors · 2023-11-16
>
> Thanks for your review. Some common questions are answered in the response overview above. Here we provide answers to the specific questions you raised.
>
> > Although the paper proposes a good concept for program verification, it's still in a nascent stage and has a long way to go for real-world deployment. I understand that this is a first step towards a bigger objective when the author mentions that "we predict that Clover, coupled with steadily improving generative AI and formal tools, will enable a future in which fully automatic, scalable generation of formally verified code is feasible. This paper charts the first steps towards realizing this vision". However, a few aspects of the paper concern me. For example, the Clover paradigm is tested on a simple and very small dataset of 60 programs that contain at most one helper function (or even none). This is a significantly trivial benchmark to test on, even for a proof of concept. In my opinion, for the readers to be convinced of the applicability of the Clover paradigm, it should be tested on relatively larger benchmarks.
>
> Thank you for your interest in our direction of work and valid suggestions regarding how we might proceed next.  Please refer to our general response to Question1.
>
> Regarding your concerns on CloverBench dataset, it is under active development and currently has 60+ examples.
> What you see in the 60 examples is the necessary and unavoidable atomic components of larger benchmarks. Most frameworks start from intraprocedural analysis and then extend to interprocedural analysis. We do have plans to expand the benchmarks to be interprocedural. Please refer to our general response on Verus results. It shows that Clover can be multilingual and still has good performance.
>
> May we ask which aspects of the dataset would you be most interested in seeing an extension of?
>
> > Even for a proof of concept, a paper should argue how it can suffice for challenging cases. When a paper aims to propose a promising program verification approach, it is not sufficient to say that "This test is, of course, imprecise (complete but not sound), but our evaluation suggests that it suffices for the level of complexity in CloverBench. More advanced equivalence checking techniques might be required for more complex examples." or "This method is conservative in the sense that it succeeds only if the two annotations are indeed equivalent, but it may fail on equivalent annotations due to limitations of the verification tool being used.". It would be nice to get an insight into how the authors aim to check (or at least approximately) code equivalence for difficult benchmarks.
>
> There are well-established formal techniques for checking that two pieces of code are equivalent.  For example, the literature on translation validation contains algorithms for checking the equivalence of two pieces of code before and after optimization.  We expect that such techniques can be leveraged for Clover’s equivalence checking.  We will clarify this roadmap in the final version.
>
> > The paper provides no insight into the performance of Clover on other mainstream programming languages like C, C++, Java, and Python. Although I agree with the fact that AI-generated codes should include formal specifications and natural language descriptions which is inherent in a language like Dafny that supports formal specification, there should be at least some insight about how this should be tackled in or extended to the mainstream languages of today. Otherwise, this would essentially mean that to get the benefit of a closed-loop verifier for AI-generated codes, the whole community should shift towards a verifier-friendly language like Dafny, which is not possible.
>
> Please refer to our general response to Question 3.
>
> The ability to use Clover on other languages depends, not on modifications to Clover, but on the availability of verification tools.  Thus, the pivot we are hoping to help inspire is not for everyone to write their code in Dafny, but for the verification community to provide stronger tools for mainstream languages.  That said, there are some tools already, and a natural direction for future work is to explore instantiations of Clover using those tools.

---

> ### Author Response · Authors · 2023-11-16
>
> > The related work of the paper is not elaborate enough and fails to describe in which line of work the present paper is closest to. Is the paper the first to propose a verifier for (AI/LLM)-based code generation from docstrings and annotations? If not, what were the limitations in the other papers that motivated the authors to come up with this approach? The related work will also look better if structured by organizing it into comprehensive categories that encompass various code verification methods in use.
>
> We will revise the section about related work as you suggest. However, Clover does propose a new paradigm that differs significantly from any previous related work. In particular, to answer your question, yes - we believe this is the first work to suggest using consistency between docstrings, annotations, and code to ensure correctness. Some previous works explore the possibility of adding assertions to the generated code (Wu 2023), which correspond to a partial  but not a complete annotation. Other works use dynamic analyzers as the ground truth (Pei 2023) and train LLMs to generate program invariants to partially specify pre/post conditions (they do not explicitly support quantifiers). Clover is the first attempt to propose a systematic mechanism of adding verification in the loop under the complete definition of deductive program verification.
>
> Kexin Pei, David Bieber, Kensen Shi, Charles Sutton, and Pengcheng Yin. 2023. Can large language models reason about program invariants? In Proceedings of the 40th International Conference on Machine Learning (ICML'23), Vol. 202. JMLR.org, Article 1144, 27496–27520.
>
> Wu, Haoze et al. “Lemur: Integrating Large Language Models in Automated Program Verification.” ArXiv abs/2310.04870 (2023): n. pag.
>
> > The paper does not compare their proposed approach against other similar works in program verification. The input requirements for Clover are quite strong (in the sense that it requires annotation and documentation along with the code). So, an AI-generated program that has all these three can be verified by Clover and any other similar SoTA program verification approaches that do not have such strong requirements. As such, it makes sense to provide a comprehensive comparison with SoTA approaches and justify empirically whether there is a significant improvement in verification filtering after imposing the strong input requirements. If not, why will the strong input requirements make sense for an automated verifier?
>
> Clover is the first to propose a consistency checking algorithm between docstrings, annotations, and code.  Thus, there are no good targets for direct comparison.  The suggestion is to look at other SoTA program verification approaches.  Program verification typically requires the code and the specification only.  Here, Dafny is SoTA and we are using it in the loop in Clover, so I’m not sure it makes sense to compare Clover with Dafny.  What we can do is compare how the 6 checks compare with using only the Dafny check.  In addition, we have added a new dataset of Verus (Lattuada 2023), which is a rust-like language and verifier.  We could also potentially compare Dafny with other program verification systems, but since our paper is not about verification systems, this doesn’t seem to make a lot of sense.
>
> We have added End2End experiment to show that even with the strong Clover filter, we have a good acceptance rate for correct generations while maintaining the perfect rejection rate. We believe this empirically justifies there is improvement in verification filtering after imposing the strong input requirements.
>
> As mentioned, we believe that there are no fully comparable SoTA approaches out there that can do exactly what we do. But if you have a specific ablation study in mind that makes sense, we would be willing to add it. May we ask what exactly you are thinking about?
>
> Andrea Lattuada, Travis Hance, Chanhee Cho, Matthias Brun, Isitha Subasinghe, Yi Zhou, Jon Howell, Bryan Parno, and Chris Hawblitzel. 2023. Verus: Verifying Rust Programs using Linear Ghost Types. Proc. ACM Program. Lang. 7, OOPSLA1, Article 85 (April 2023), 30 pages. https://doi.org/10.1145/3586037
>
> > Although the paper is well-written, I believe the structure could be made more reader-friendly. It needed two passes for me to understand the concept. The insights given in the later part of the introduction felt too abstract on the initial read. It required me to go through the whole proposed approach and come back to the introduction again to grasp the insights given in the introduction. Algorithm-1 is helpful in this context, but I presume that was pushed to the appendix due to lack of space.
>
> Thanks for the feedback. We’ll reorganize the writing and move concrete descriptions earlier in a revised version.

---

> ### Author Response · Authors · 2023-11-16
>
> > Consider a situation: A user provides a prompt to the LLM to generate a code according to some specification (e.g. sort the input array in ascending order). The LLM generates a code that sorts the input array in descending order. Along with the code, the LLM provides annotation and docstring that are fully consistent with the generated code. In this case, the Clover checks will be successful because the code-annotation-docstring is mutually consistent with the six checks. But, the code does not match the user specifications. Because the paper proposes a closed-loop approach of code generation + verification, it would be insightful to tackle such odd but common cases where the generated annotation, docstring, and code are mutually consistent, but do not tally with the prompt given by the user to generate the code (original functionality intended by the user).
>
> Please refer to our general response to Question 4.
>
> > (Minor) There are a few spelling and syntactic mistakes in the current version e.g. "tranlations" on Page 6, "COLVER" on Page 22, and space missing before full-stops in quite a few places.
>
> Thanks! We’ll fix them in an updated version.
>
> > Reproducibility: I did not see any URL to access the code or dataset.
>
> We have attached the code/dataset in this rebuttal.
>
> > Is the paper the first to propose a verifier for (AI/LLM)-based code generation from docstrings and annotations? If not, what were the limitations in the other papers that motivated the authors to come up with this approach?
>
> Clover does propose a new paradigm that differs significantly from any previous related work. In particular, to answer your question, yes - we believe this is the first to propose a verifier for (AI/LLM)-based code generation from docstrings and annotations. Some previous works explore the possibility of adding assertions to the generated code (Wu 2023), which correspond to a partial but not a complete annotation. Other works use dynamic analyzers as the ground truth (Pei 2023) and train LLMs to generate program invariants to partially specify pre/post conditions (they do not explicitly support quantifiers). Clover is the first attempt to propose a systematic mechanism of adding verification in the loop under the complete definition of deductive program verification.
>
> Kexin Pei, David Bieber, Kensen Shi, Charles Sutton, and Pengcheng Yin. 2023. Can large language models reason about program invariants? In Proceedings of the 40th International Conference on Machine Learning (ICML'23), Vol. 202. JMLR.org, Article 1144, 27496–27520.
>
> Wu, Haoze et al. “Lemur: Integrating Large Language Models in Automated Program Verification.” ArXiv abs/2310.04870 (2023): n. pag.

---

> ### Author Response · Authors · 2023-11-21
>
> Dear reviewer,
>
> We are really encouraged by your helpful and positive feedback and sincerely appreciate your efforts. We hope our answer, updated experiments, and the corresponding manuscript make the paper clearer and stronger. Please let us know if you have any further comments, and we are more than happy to address them.
>
> Best,
>
> Submission113 authors

---

### Official Review · Reviewer_kgwt · 2023-11-04

**Soundness:** 3 good
**Presentation:** 2 fair
**Contribution:** 2 fair
**Rating:** 6
**Confidence:** 4

**Summary:**

This paper presents Clover, a method to generate safe and formally verified code using LLMs.
In the first step, Clover can utilize LLMs like GPT-4 to generate (i) Code (ii) DocStrings in Natural Language (iii) Formal Specifications (e.g. loop-invariants, pre and post conditions).
In the second step, Clover applies pairwise consistency checks on the LLM generated Code, Docstrings, and Specifications.
More specifically, consistency checks are performed via reconstruction testing (for Specifications <=> Docstring and Code <=> Docstring checks) and deductive verification tools (for Code <=> Specification checks). Reconstruction testing utilizes LLMs to reconstruct one component (e.g. Code ) from another (e.g. Docstring), and further uses an LLM to verify the equivalence between the reconstructed and the original output.

The paper makes reasonable contributions overall, and its strengths and contributions include choosing and formalizing an important problem, and dataset (CloverBench) construction.

**Strengths:**

- This paper explores a method to rigourously verify the artifacts generated by Code-LLMs. This problem should be of wide interest, but there seem to be little efforts in this direction.
- Clover exhibits high acceptance rates for correctly generated artifacts while precisely rejecting the incorrect ones.
- The paper contributes CloverBench, which could be a useful benchmark in the future.

**Weaknesses:**

- It seems Clover will need to depend on very strong LLMs (e.g., to generate code/docstring from just the annotations). Thus, consistency checks may often fail if the LLM is weak.
- Dependence on a verification tool. (Can it check any given java file or just short program snippets?)
- A very small evaluation set of just 60 examples.

**Questions:**

- How much do code LLMs (e.g., CodeLlama, StarCoder, WizardCoder) benifit from the Clover approach? What fraction of the code generated by these LLMs is typically unacceptable in the first place?
- It would be interesting to see experiments with open-source LLMs as well.
- How does Clover ensure the security aspects of the generated code? Is there an actual example in the eval set (CloverBench) to demonstrate it?
- What is the domain of dafny examples (competitive programming / ... ?)
- In Section 4.2, how do you determine whether GPT-4 passes or fails for the task of generating Code/Annotations? Do you use Clover to determine pass/fail?
- Can Clover be extended to repository level code verification?

---

> ### Author Response · Authors · 2023-11-16
>
> Thank you for your questions and comments! We will try to answer all of them. Feel free to add more questions/comments.
>
> > It seems Clover will need to depend on very strong LLMs (e.g., to generate code/docstring from just the annotations). Thus, consistency checks may often fail if the LLM is weak.
>
> It is true that the overall Clover success rate depends on the specific tools in each component, especially LLMs used for generation, as do most frameworks. Arguably this is not a Clover weakness. We are not aiming to train/fine-tune a generation model, although this could be a future direction. One interesting idea we have in mind is to fine-tune an LLM to achieve annotation domain knowledge transfer. In this way, with a fine-tuned model for generating pre/post conditions and loop invariants, even less mainstream annotation languages can be good candidates for the Clover framework.
> Our experimental results support the hypothesis that LLMs are strong enough for this task.  With k=10, we only fail 13% of the time.  And future efforts can improve this rate further.
>
> > Dependence on a verification tool. (Can it check any given java file or just short program snippets?)
>
> Please refer to our general response to Question 3 for information about why we use Dafny and its capabilities  Formal verification is a rich field with a long history.  Over time, the capabilities of formal tools have improved dramatically.  Today, tools like Dafny can be used to verify very large programs, as long as they are annotated with formal specifications.
>
> For Java code specifically, there are a limited number of formal tools.  Perhaps the most prominent is the Java Pathfinder (JPF) system, a tool developed by NASA to verify executable Java bytecode programs. It employs primarily symbolic execution techniques. The main application has been Model checking of concurrent programs, to find defects such as data races and deadlocks.  Clover requires a specific kind of verification tool, one that supports deductive verification.  JPF is not designed as a deductive verification framework though it could be used to help construct one.  We hope that our work will help motivate such efforts.
>
> > A very small evaluation set of just 60 examples.
>
> Please refer to our general response to Question 1.
>
> > How much do code LLMs (e.g., CodeLlama, StarCoder, WizardCoder) benifit from the Clover approach? What fraction of the code generated by these LLMs is typically unacceptable in the first place?
>
> We have experimented with CodeLlama and the results are shown in the general response. To answer the second question, please see our general comment on End2End experiment.
>
> > It would be interesting to see experiments with open-source LLMs as well.
>
> Please see our general comment on CodeLlama.
>
> > How does Clover ensure the security aspects of the generated code? Is there an actual example in the eval set (CloverBench) to demonstrate it?
>
> When security properties can be formulated as functional properties of the code, Clover can check them.  Moreover, examples written in Dafny can ensure no side effects (no illegal memory access, heap safety, etc.) happen during memory read/write operations.
>
> The ```BubbleSort``` example in CloverBench is an example of this. Not only can we ensure the functional correctness (ascending order and all elements remain the same), we are able to make sure that the memory operations are confined to the input array a with ```modifies a```. Code can only access memory locations specified after ```modifies``` clause.
> ```
> method BubbleSort(a: array<int>)
>  modifies a
>  ensures forall i,j::0<= i < j < a.Length ==> a[i] <= a[j]
>  ensures multiset(a[..])==multiset(old(a[..]))
> {
>  var i := a.Length - 1;
>  while (i > 0)
>    invariant i < 0 ==> a.Length == 0
>    invariant -1 <= i < a.Length
>    invariant forall ii,jj::i <= ii< jj <a.Length ==> a[ii] <= a[jj]
>    invariant forall k,k'::0<=k<=i<k'<a.Length==>a[k]<=a[k']
>    invariant multiset(a[..])==multiset(old(a[..]))
>  {
>    var j := 0;
>    while (j < i)
>      invariant 0 < i < a.Length && 0 <= j <= i
>      invariant forall ii,jj::i<= ii <= jj <a.Length ==> a[ii] <= a[jj]
>      invariant forall k, k'::0<=k<=i<k'<a.Length==>a[k]<=a[k']
>      invariant forall k :: 0 <= k <= j ==> a[k] <= a[j]
>      invariant multiset(a[..])==multiset(old(a[..]))
>    {
>      if (a[j] > a[j + 1])
>      {
>        a[j], a[j + 1] := a[j + 1], a[j];
>      }
>      j := j + 1;
>    }
>    i := i - 1;
>  }
> }
> ```

---

> ### Author Response · Authors · 2023-11-16
>
> > What is the domain of dafny examples (competitive programming / ... ?)
>
> Textbook algorithms that you may find in undergraduate introductory algorithm/data structures classes.
>
> > In Section 4.2, how do you determine whether GPT-4 passes or fails for the task of generating Code/Annotations? Do you use Clover to determine pass/fail?
>
> We did not use Clover in Section 4.2 as doing so would make it unclear whether problems are in the generation phase or the checking phase, and we want to focus on only the generation phase.  Thus, checking for Section 4.2 was done manually.
>
> > Can Clover be extended to repository level code verification?
>
> Please refer to our general response to Question 1.
>
> Yes. That is our vision and we will move in that direction in our next project. For larger code bases, we will need to consider:
> 1. Interprocedural verification protocols.  We have an idea for this that we call CloverChain.  As mentioned in our general response, we will sketch this idea in the future work section.
> 2. Support for complex language syntax and large LLM context windows
> 3. Advanced equivalence checking methods (e.g., formal equivalence checking for code)

---

> ### Author Response · Authors · 2023-11-21
>
> Dear reviewer,
>
> We are really encouraged by your helpful and positive feedback and sincerely appreciate your efforts. We hope our answer, updated experiments, and the corresponding manuscript make the paper clearer and stronger. Please let us know if you have any further comments, and we are more than happy to address them.
>
> Best,
>
> Submission113 authors

---

### Official Review · Reviewer_M5ic · 2023-11-10

**Soundness:** 2 fair
**Presentation:** 2 fair
**Contribution:** 2 fair
**Rating:** 3
**Confidence:** 4

**Summary:**

This paper proposes Clover to check the consistency among code, docstrings and formal annotations. Specifically, it conducts a six-way check: code to docstring, docstring to code, code to annotation, annotation to code, docstring to annotation, and annotation to code. Clovers does experiment with a benchmark of 60 small programs.

Clover uses a verification tool to check whether code satisfies the annotation, which I agree. But I have a concern. The programs used in the paper are simple ones. As far as I know, verification tools suffer from scalability issues. I would like to know the insights of the authors regarding this.

To check whether an annotation is consistent with code, Clover uses LLM to generate new code for a given annotation and checks whether the generated code and the original are equivalent. Firstly, in many cases, there would be multiple annotations for a given piece of code. For example, for a sorting algorithms, an annotation can specify all the elements in a given set are in ascending order and another annotation specifies all the elements in the given set remain the same. How Clover deal with such cases? Secondly, the generated code from LLM may not be consistent with the given annotation. So if the generated code and the original code are inconsistent, it does not mean the annotation does not align with the original code. This second concern also holds for other checking. Basically, Clover assumes that LLM must have a very high accuracy in translating one artifact to another. I admit that LLMs are powerful, but LLMs do have limitations like hallucinations. I would suggest to discuss the threats to validity. Moreover, the paper claims that Clover has the potential to improve the trustworthiness of code generated by LLMs. It is kind of the chicken-egg thing.

When checking whether two pieces of code are consistent, Clover leverages a set of input-output pairs which makes a lot of sense. But such pairs are not always availble. What would be the solution without the pairs? Sometimes, we do have such data availble, but not complete. It is possible that the two pieces of code pass the test, but actually they are inconsistent. I would suggest the paper includes some disscussion on this.

The paper employs LLM to check the semantic equivalence between two natural language sentenses. Natural languages are ambigious. Even human may not fully understand the actual semantic in a natural language sentence. I have concerns on the acccuracy of the result from LLM. Also, the paper does not have details regarding how to use LLMs to do this task.

Clover is used on a very small dataset of 60 textbook-style programs. I would suggest the authors to test on relatively larger benchmarks to better demonstrate the potencial of the proposed work.

**Strengths:**

+ An important problem
+ Easy follow paper

**Weaknesses:**

- Unsound design
- Small-scale experiment
- Lack important details

**Questions:**

How LLMs are used to check whether two docstrings are semantically equivalent?

---

> ### Author Response · Authors · 2023-11-16
>
> Thank you for your questions and comments! We will try to answer all of them. Feel free to add more questions/comments.
>
> > The programs used in the paper are simple ones. As far as I know, verification tools suffer from scalability issues. I would like to know the insights of the authors regarding this.
>
> Answered in General Response 1.
>
> > Firstly, in many cases, there would be multiple annotations for a given piece of code. For example, for a sorting algorithms, an annotation can specify all the elements in a given set are in ascending order and another annotation specifies all the elements in the given set remain the same. How Clover deal with such cases?
>
> When we say “annotation,” we mean the union of all of the individual logical specification formulas, including preconditions, postconditions, and loop invariants. It is quite possible for an “annotation” to include more than one postcondition.  We will clarify this in the final version.
> Note that the key requirement is that the annotation is sufficiently precise to uniquely determine the output for any given input.  It does not matter how many pre-/post-conditions are required to achieve this. We have specified in the paper that the Clover input constraints restrict the input docstring, code, and annotation triplets to define one and only one equivalent class of output (formally defined in Section 3.2). In other words, it expects that each of the three components provides sufficient detail to unambiguously determine a unique result of running the code on any given input.
>
> > Secondly, the generated code from LLM may not be consistent with the given annotation. So if the generated code and the original code are inconsistent, it does not mean the annotation does not align with the original code. This second concern also holds for other checking.
>
> We are assuming you are referring to the reconstruction step when verifying the consistency between code and annotation.  And you are asking whether it could happen that the code and annotation are in fact consistent, but we get a false negative because the LLM generates code from the annotation that is not consistent with it.
>
> Yes, this can happen and could lead to false negatives (we reject a consistent triple as inconsistent).  Note that we in fact bias our entire framework towards false negatives, because we want to avoid a false positive at all costs.  Our empirical results show that in this we were quite successful, as we never had a single false positive.  However, even with all of the possibilities for false negatives, we still only get false negatives 25% of the time.  If we use k=10, this goes down to 13%.  One of the promising and perhaps surprising results of the paper is that despite the potential for such false negatives, the scheme works pretty well for a first attempt.  And there are many avenues available to improve things going forward.
>
> > Basically, Clover assumes that LLM must have a very high accuracy in translating one artifact to another. I admit that LLMs are powerful, but LLMs do have limitations like hallucinations. I would suggest to discuss the threats to validity.
>
> Clover relies on LLM’s being good at translation, but does not assume that they are.  This is a subtle difference.  The consistency checking is designed in such a way that if any LLM fails because of hallucinations, then the entire consistency check fails.  As mentioned above, this biases the system towards false negatives while providing strong protection against false positives.
>
> The community will not stop using LLMs because they can hallucinate, nor do we believe there exists the need to. But when it comes to security-critical high-stakes code generation, it would be reassuring to have strong guarantees about code correctness. This is what Clover is designed to do. If hallucinations appear in the generation phase, the generated artifacts have a high probability of being rejected by Clover’s six consistency checks. The presence of a formal tool (the Dafny verifier) ensures that certain kinds of inconsistencies are always rejected, protecting against those kinds of hallucinations. Hallucinations can also happen during reconstruction steps, which could affect the accuracy of the Clover consistency checking. However, we believe that the chances of a false positive are very low.  Our experiments show a high acceptance rate for ground-truth examples and a 100% reject rate for incorrect examples, providing evidence for this claim. The intuition behind this goes as follows: since we have 6 separate checks, we believe that an inconsistent triple has a low probability of succeeding on all six checks accidentally. At the same time, with several independent tries, a reasonable though not perfect success rate for reconstruction delivers a good pass rate for correct ones.

---

> ### Author Response · Authors · 2023-11-16
>
> > Moreover, the paper claims that Clover has the potential to improve the trustworthiness of code generated by LLMs. It is kind of the chicken-egg thing.
>
> This paper is not about improving  LLM code generation capabilities. It is about automatically determining whether generated code is correct and should be trusted. While it is true that we rely on some generation capabilities for this check, we bias these towards false negatives and we also rely on formal methods for some checks.  The result is that even though LLMs themselves are not trustworthy, our consistency check is highly trustworthy.  In fact, it never accepts incorrect triples in our experiments.  It is similar to how redundant or fault-tolerant computing can produce highly reliable systems even though individual pieces are not reliable.
>
> > When checking whether two pieces of code are consistent, Clover leverages a set of input-output pairs which makes a lot of sense. But such pairs are not always availble. What would be the solution without the pairs? Sometimes, we do have such data availble, but not complete. It is possible that the two pieces of code pass the test, but actually they are inconsistent.
>
> This is a valid point. Sometimes unit tests do not exist in the wild. And unit tests are inherently incomplete. There are other more advanced alternatives, including concolic/symbolic testing and formal equivalence checking, but these are overkill for our simple examples.
> In future work, as we tackle more challenging examples, we plan to bring in these more advanced alternatives.  These can also be used in the case when no unit tests exist.
>
>
> > The paper employs LLM to check the semantic equivalence between two natural language sentenses. Natural languages are ambigious. Even human may not fully understand the actual semantic in a natural language sentence. I have concerns on the acccuracy of the result from LLM. Also, the paper does not have details regarding how to use LLMs to do this task.
>
> Please refer to our general response to docstring equivalence checking in Q2.
>
>
> > Clover is used on a very small dataset of 60 textbook-style programs. I would suggest the authors to test on relatively larger benchmarks to better demonstrate the potencial of the proposed work.
>
> This is definitely a great suggestion. Please refer to our general response to CloverBench dataset and how we plan to scale.  As we argue there, it is important to start simple in order to test the first few hypotheses and not try to jump too far too soon.  We believe this work validates our initial hypotheses and provides a good baseline for scaling up to large interprocedural analysis.
> CloverBench is a long-term project, and we will continue to improve it over time.  Note that it is designed manually, and each benchmark requires hand-written formal annotations and  multiple variants, so it is some work to scale up the benchmark set.
>
> > How LLMs are used to check whether two docstrings are semantically equivalent?
>
> After playing with different embedding models, we finally choose to ask GPT4 if two docstrings are semantically equivalent. The concrete prompt is in the appendix (page 20).

---

> ### Author Response · Authors · 2023-11-21
>
> Dear reviewer,
>
> We are really encouraged by your helpful and positive feedback and sincerely appreciate your efforts. We hope our answer, updated experiments, and the corresponding manuscript make the paper clearer and stronger. Please let us know if you have any further comments, and we are more than happy to address them.
>
> Best,
>
> Submission113 authors

---

### Author Response · Authors · 2023-11-16
**General Response to Common Questions**

We want to thank all the reviewers for their insightful comments and suggestions!
We address common concerns in this general reply and will provide answers to specific questions in the individual replies.

**Q1: The current dataset is small, and the examples are simple. How does it prove the Clover paradigm can scale to real systems? How could it generalize to other languages?**

The Clover vision aims to eventually solve large problems. But a lot of hypotheses need to be tested to eventually achieve that vision. This paper answers two important and fundamental questions: Is it possible to generate annotations? Does consistency checking work for checking code correctness? We are answering the most crucial questions first. To test those, we start with simple examples written in Dafny. Even in real distributed systems like IronFleet (https://github.com/microsoft/Ironclad/tree/main/ironfleet) implemented in Dafny, each procedure will not be much larger than the ones we have. As opposed to other formal verification techniques, deductive verification does not typically have scalability issues, because each query represents a verification condition covering a few lines of code.  Thus, each query to the underlying solver is small.  Large programs will have more queries, but if each query is small, then verification is scalable. **Deductive verification is rather bottlenecked by writing specifications (addressed by the Clover generation phase), not verification.** A real system might contain thousands of functions, but each function could still be small (10-30’s of LOC). Therefore, we think CloverBench is reasonable as a proof of concept.

To reach the eventual vision of having Clover working on real large systems, there are two steps: (1) show Clover works on for intraprocedural analysis, (2) show Clover can be extended to interprocedural analysis.
This work focuses on showing step (1), and step (2) is left for future work (the next paper). This is in line with standard approaches in Program Languages research. We do have plans to show how to achieve step (2). We plan to decompose complex procedure calls into individual self-contained smaller procedure calls. Then we plan to verify the decomposed calls layer by layer and thus reduce to step (1). We will be happy to sketch this plan in the future work section of the paper.

**Q2: Can Clover work on other languages? Why start with Dafny?**

Clover relies on a strong deductive verification back-end.  Dafny is one of the most developed and well-maintained deductive program verification languages. Verus (Lattuada et al., 2023) is another similar language but only came out in 2023.
The Clover paradigm is orthogonal to the choice of programming language. It can definitely be applied to more mainstream languages such as Python, Java, C, etc. However, the verification tools for those languages are not as advanced as they are for Dafny, which was designed with verification in mind.

Moreover, there are a number of real systems implemented in Dafny.  See, for example:
git@github.com:microsoft/Armada.git
git@github.com:aws/aws-database-encryption-sdk-dynamodb-java.git
git@github.com:aws/aws-encryption-sdk-dafny.git
git@github.com:cedar-policy/cedar-spec.git
git@github.com:mit-pdos/daisy-nfsd.git
git@github.com:Consensys/evm-dafny.git
git@github.com:microsoft/Ironclad.git
git@github.com:vmware-labs/verified-betrfs.git
git@github.com:project-everest/vale.git
git@github.com:secure-foundations/veribetrkv-osdi2020.git
https://github.com/vmware-labs/verified-betrfs/tree/splinter
https://github.com/secure-foundations/everquic-dafny
Given this, we feel that our choice of Dafny language is reasonable, as our first goal is to demonstrate the feasibility of Clover as a paradigm, not to build tools for a specific programming language.
In future work, we would also like to demonstrate Clover in other languages for broader impact. However, that will require finding or building good deductive verification solutions for those languages and are weakly related to the proof of the Clover concept.

References:
[1] Lattuada, Andrea, et al. “Verus: Verifying Rust Programs using Linear Ghost Types.” PACMPL 2023

---

> ### Author Response · Authors · 2023-11-16
>
> **Q3: Equivalence checking is always a challenge. This challenge is even more pronounced for checking "equivalence" between natural language descriptions. How does the writer implement docstring equivalence check?**
>
> Our goal in this paper is to present a general framework that can be improved over time. We believe that including docstrings makes the framework more general and more powerful.  At the same time, we acknowledge that docstring equivalence checking is currently weak and identify this as an area of future research that can improve the efficacy of Clover. We have included the details on how we ask GPT to perform the task, and the concrete prompt we sent to GPT4 is included in the appendix (page 20). We show empirically that the six checks are surprisingly robust even with imperfect equivalence checks.
>
> **Q4: What if LLM hallucinates and is consistently wrong? What if some user asks in natural language to “sort the input array in ascending order”, but the LLM generates a docstring, code, and annotation triplet in descending order?**
>
> In phase one (generation), at least one of the docstring, code, and annotation is provided by a human. The LLM then generates the others, but crucially, it is not allowed to change the provided piece. We will clarify this in the final version. So, in the proposed situation, “sort the input array in ascending order,” we would use this as the docstring and not allow the LLM to change it. We are then guaranteed that if the code and annotation are consistent with the docstring, then this is solving the right problem. If with the ascending order in the docstring, and descending property in the code and annotation, Clover will reject it, which is demonstrated in one of our wrong variants in CloverBench.
>
> Another possible question you are raising here could be the user is asking for a task in an informal natural language description, and the LLMs need to generate a formal docstring. This is another step of prompt engineering question, which is not in the scope of this paper. But the case of "ascending" and "descending" is unlikely to happen in GPT4, as those two directly contradict each other. If the user's prompt is not specific enough, and the generated artifacts are not consistent with the human intention in their mind, it is not a solvable problem in definition. The best way to let LLMs know what a user wants is better alignment tuning and letting them interact with humans.
>
> **Q5:  Dafny syntax mistakes get in the way of reconstruction testing.**
>
> Even though Dafny is not as mainstream as languages like Python or Java, most of the Dafny syntax mistakes can often be fixed by having the Dafny compiler give feedback to GPT4. For example, on its first try, GPT4 might provide an annotation that modifies an array without using the “modifies” keyword required by Dafny.  After feedback from the compiler, GPT4 does provide the correct keyword. Some problems are hard to fix. Examples include (1) GPT4 may use native Dafny data structures in the wrong way. For example, the native Dafny map data structure is initialized as ```var r := map[]``` not ```var r:=map{}```. (2) Another prominent case is GPT4’s lack of understanding of ghost variables which are specifically designed for verification purposes (not used in program execution and ignored by compilers). The use of ghost variables is subtle and creates context limitations for other variables. We observe empirically that GPT4 is unable to use ghost variables correctly most of the time. (3) Triggers are also difficult to get right. A trigger is another advanced Dafny feature used to accelerate the verification process when specifications include logical quantifiers. (4) functions parameterized on types are also hard to get right since their use differs slightly from predefined types of functions/data structures.
>
> The good news is that even with these limitations, reconstruction succeeds in finding the correct Dafny syntax most of the time.
> Empirically, in the k=10 experiment, out of the 8 failure cases in accepting ground truth examples, only 3 were due to fundamental Dafny syntax generation failures.  This is also a low-hanging fruit for future efforts to improve Clover, as fine-tuning on Dafny syntax should be able to solve many of those issues.

---

### Author Response · Authors · 2023-11-18
**New experiment results on Verus**

We have added 10 examples written in Verus (A language for verifying Rust) and run the Clover consistency checks. The dataset has been updated in the supplement.  With maximal feedback from Verus/Rustc compiler (same setting as k=1 in the paper), it is able to accept 8 out of 10 true examples. Here is the detailed result:


|                       | anno_sound    | anno_complete | code2doc  | doc2code|anno2doc |  doc2anno |
| ---------------- | --------- | -------- | --------- |---------------- | --------- | -------- |
| abs   | A |A |A|A | A | A |
| binary_search | A |A |A|A | A | R |
| is_prime      | A |R |A|A | A | R |
| max | A |A |A|A | A | A |
| min   | A |A |A|A | A | A |
| min3   | A |A |A|A | A | A |
| pop   | A |A |A|A | A | A |
| push   | A |A |A|A | A | A |
| return_2   | A |A |A|A | A | A |
| reverse   | A |A |A|A | A | A |

---

### Author Response · Authors · 2023-11-21
**Experimental results on CodeLlama-34B**

We present the results of CodeLlama-34b on Clover tests. The results indicate that CodeLlama-34b is incapable of dealing with Dafny code, neither verification logic. In the early stages of this project, we briefly experimented with several other open models and discovered that most are significantly lacking in their ability to handle tasks related to program verification. While many open models have shown impressive results in other popular datasets, our dataset offers a perspective on the extent of the gap in knowledge coverage between open models and GPT-4. Specifically, open models perform notably poorly with low-resource languages. But GPT-4 highlights the potential of AI in verification tasks. Therefore, one of our goals is to showcase its promise and to garner more attention towards integrating verification into the workflow from both AI and verification specialists.

| | anno_sound  | anno_complete | code2doc| doc2code| anno2doc|doc2anno | Clover 6 edges |
| ---------------- | --------- | -------- | --------- |---------------- | --------- | -------- |-----|
| ground truth |      60/60       |       6/60      |   8/60        |     2/60   |     48/60  |  2/60    | 2/60    |

---

### Author Response · Authors · 2023-11-21
**Experiments on End2End**

We have included some end-to-end experiments (encompassing both generation and verification phases) to demonstrate how Clover can assist with generation. It is important to note that Clover's application and motivation extend beyond aiding code generation from docstrings; it is also useful in facilitating program verification. Program verification is a technique already utilized in many applications. Clover's mission is to alleviate the scalability issue of manually writing specifications by enabling automatic generation and correctness checking.

The table below shows the results from generating docstring and code from annotation alone. For each example, we did 200 independent generations. Then we performed 6 Clover consistency checks and noted the ones being accepted. The results show a good acceptance rate for correct generations and a perfect rejection rate for wrong generations. This demonstrates that Clover not only works on manually curated examples but also works on LLM-generated ones.

| GenFromAnno | correct | correct & accept | correct & reject | incorrect & accept |
|----|----|----|----|---|
|abs| 189  | 185 | 4 | 0 |
|array_sum| 111 |54  | 57 | 0 |
|cal_sum| 190  | 183| 7|0 |
|double_quadruple| 195  | 190|5 | 0 |
|is_even | 191  | 184| 7 | 0 |
|swap_arith| 118  | 114| 4 | 0 |
|swap_sim|198  | 194|4 | 0 |
|triple2| 189  | 184|  5 | 0 |



This table shows the results from generating annotation and code from docstring alone.
| GenFromDoc |  correct | correct & accept | correct & reject | incorrect & accept |
|----|----|----|----|---|
|array_append| 104        | 89                                    | 15                     | 0                            |
|array_product| 102        | 74                                        | 28                     | 0                            |
|cal_ans|138        | 135                                            | 3                     | 0                            |
|compare| 191        | 185                                     | 6                     | 0                            |
|find| 117        | 87                                    | 30                     | 0                            |
|linear_search1| 133        | 74                               | 59                     | 0                            |
|min_of_two| 198        | 196                             | 2                     | 0                            |
|remove_front| 84        | 64                                |20                     | 0                            |
|swap| 184        | 180                               | 4                   | 0                            |
|swap_in_array| 184        | 174                                    | 10                     | 0                            |
|triple| 197  |197                          | 0                    | 0                            |
|triple4| 199        | 199                            | 0                    | 0                            |


This table shows the results from generating docstring and annotation from code alone.

|  GenFromCode|correct | correct & accept | correct & reject | incorrect & accept |
|----|----|----|----|---|
|all_digits|151        | 144                                     | 7                    | 0                            |
|array_copy|191        | 174                                     | 17                    | 0                            |
|avg|177        | 174                                    | 3                   | 0                            |
|insert| 162        | 101                                   | 61                    | 0                            |
|min_array| 180        | 111                                 | 69                    | 0                            |
|multi_return| 174        | 102                     | 72                     | 0                            |
|quotient| 155        | 94                            | 61                     | 0                            |
|return_seven| 187        | 160                         | 27                 | 0                            |
|swap_bitvector| 158        | 145                          | 13                   | 0                            |
|test_array| 132        | 105                        | 27                  | 0                            |
|triple3| 192        | 190                        | 2                 | 0                            |
|update_array| 113        | 68                   | 45                  | 0                            |